# Exploring the processes controlling secondary inorganic aerosol: Evaluating the global GEOS-Chem simulation using a suite of aircraft campaigns

Olivia G. Norman[1], Colette L. Heald[1,2,a], Solomon Bililign[3], Pedro Campuzano-Jost[4], Hugh Coe[5,6], Marc N. Fiddler[7], Jaime R. Green[8], Jose L. Jimenez[4], Katharina Kaiser[9], Jin Liao[10,11], Ann M. Middlebrook[12], Benjamin A. Nault[4,13,14], John B. Nowak[15], Johannes Schneider[9], André Welti[16]

[1] Department of Earth, Atmospheric and Planetary Sciences, Massachusetts Institute of Technology, Cambridge, MA, USA
[2] Department of Civil and Environmental Engineering, Massachusetts Institute of Technology, Cambridge, MA, USA
[3] Department of Physics, North Carolina Agricultural and Technical State University, Greensboro, North Carolina
[4] Department of Chemistry and Cooperative Institute for Research in Environmental Science (CIRES), University of Colorado, Boulder, CO, USA
[5] University of Manchester, Oxford Road, Manchester, M13 1QD, UK
[6] National Centre for Atmospheric Sciences, University of Manchester, Oxford Road, Manchester, M13 1QD, UK
[7] Department of Chemistry, North Carolina Agricultural and Technical State University, Greensboro, NC, USA
[8] Department of Environmental Sciences & Engineering, University of North Carolina, Chapel Hill, NC, USA
[9] Particle Chemistry Department, Max Planck Institute for Chemistry, Mainz, Germany
[10] NASA Goddard Space Flight Center, Greenbelt, MD, USA
[11] Goddard Earth Sciences Technology and Research (GESTAR) II, University of Maryland, College Park, MD, USA
[12] NOAA Chemical Sciences Laboratory, Boulder, Colorado, United States
[13] Department of Environmental Health and Engineering, Johns Hopkins University, Baltimore, MD, USA
[14] Center for Aerosol and Cloud Chemistry, Aerodyne Research, Inc., Billerica, MA, USA
[15] NASA Langley Research Center, Hampton, Virginia, United States
[16] Finnish Meteorological Institute, Helsinki, Finland
[a] Now at: Department of Environmental Systems Science, ETH Zurich, Zurich, Switzerland

*Correspondence to*: Olivia G. Norman (onorman@mit.edu), Colette L. Heald (colette.heald@env.ethz.ch)

**Abstract.** Secondary inorganic aerosols (sulfate, nitrate, and ammonium; SNA) are major contributors to fine particulate matter. Predicting concentrations of these species is complicated by the cascade of processes that control their abundance, including emissions, chemistry, thermodynamic partitioning, and removal. In this study, we use 11 flight campaigns to evaluate the GEOS-Chem model performance for SNA. Across all the campaigns, the model performance is best for sulfate ($R^2 = 0.51$, NMB = 0.11) and worst for nitrate ($R^2 = 0.22$, NMB = 1.76), indicating substantive model deficiencies in the nitrate simulation. Thermodynamic partitioning reproduces the total particulate nitrate well ($R^2 = 0.79$ and NMB = 0.09), but actual partitioning (i.e., $\varepsilon(NO_3^-) = NO_3^-/TNO_3$) is challenging to assess given the limited sets of full gas and particle phase observations needed for ISORROPIA II. In particular ammonia

observations are not often included in aircraft campaigns and more routine measurements would help constrain sources of SNA model bias. Model performance is sensitive to changes in emissions and dry and wet deposition, with modest improvements associated with the inclusion of different chemical loss and production pathways (i.e., acid uptake on dust, $N_2O_5$ uptake, and $NO_3^-$ photolysis). However, these sensitivity tests show only modest reduction in the nitrate bias, with no improvement to the model skill (i.e., $R^2$) implying that more work is needed to improve the description of loss and production of nitrate and SNA as a whole.

## 1 Introduction

Aerosols (also known as particulate matter, PM) in our atmosphere are associated with poor air quality (Malm et al., 2000) and the attendant elevated risk of human premature mortality (Pope and Dockery, 2006; Huang et al., 2012), as well as changes in our climate (Lohmann and Feichter, 2005; Myhre et al., 2013). A major component of fine particulate matter ($PM_{2.5}$) is secondary inorganic aerosols, which include sulfate ($SO_4^{2-}$), nitrate ($NO_3^-$), and ammonium ($NH_4^+$). While other inorganic species, such as chloride ($Cl^-$) can be locally important (Haskins et al., 2018; Gani et al., 2019), sulfate, nitrate, and ammonium (hereafter SNA) are the dominant contributors to secondary inorganic fine aerosol worldwide, contributing between ⅓ and ¾ to measured fine non-refractory PM (Zhang et al., 2007). These inorganic aerosols have been the major aerosol constituent responsible for the degradation of air quality associated with industrialization (e.g., in the United States and Europe in the 1970s and 1980s, and China in the early 2000s), as well as subsequent improvements with the implementation of emissions control technology (Leibensperger et al., 2012; Geng et al., 2017). SNA are also the principal agents of historical aerosol climate forcing (IPCC, 2021). SNA themselves are not directly emitted, but instead are formed in the atmosphere from precursor gases that have a range of natural and anthropogenic sources. However, connecting the response of SNA concentrations to changes in the emissions can be challenging because many non-emission-related processes affect these aerosols (e.g., chemical oxidation, thermodynamic partitioning, wet and dry deposition; Pye et al., 2009; Paulot et al., 2017; Shah et al., 2018; Li et al., 2021; Zhai et al., 2021a). Understanding these formation and loss processes is key to characterizing aerosol trends and impacts on a global scale.

Emissions of sulfur dioxide ($SO_2$), nitrogen oxides ($NO_x$), and ammonia ($NH_3$) provide the source for sulfate, nitrate, and ammonium aerosols. $SO_2$ and $NO_x$ emissions are dominated by fossil fuel combustion. The major sources of $NH_3$ are agricultural emissions, originating from livestock and fertilizer use, and, in urban areas, from vehicular emissions (e.g., Phan et al., 2013; Sun et al., 2017). Other important sources include volcanoes and the oxidation of oceanic dimethyl sulfide (for $SO_2$), soils and biomass burning (for $NH_3$ and $NO_x$), and lightning (for $NO_x$). Upon emission, $SO_2$ is oxidized in both the gas- and aqueous-phase to form acidic sulfate aerosols. Similarly, the formation of inorganic nitrate is mainly through the oxidation of $NO_x$ into nitric acid ($HNO_3$; Alexander et al., 2009). The very low saturation vapor pressure of sulfuric acid implies that this species is primarily found in the particle phase (Seinfeld and Pandis, 2016). In contrast, thermodynamic partitioning controls the amount of nitrate and ammonium in the gas and particle phase (i.e., between $HNO_3$ and $NO_3^-$ for nitrate and $NH_3$ and $NH_4^+$ for ammonium). This partitioning is dependent on relative humidity, temperature, and pH, where higher relative humidity, lower temperature, and higher

aerosol pH favors nitrate partitioning into the particle phase (Fountoukis and Nenes, 2007; Guo et al., 2016). Ammonia reacts with both acidic sulfate aerosols (to form different salts, e.g., ammonium bisulfate, ammonium sulfate) and nitrate (to form particulate ammonium nitrate; Seinfeld and Pandis, 2016). VOCs can also act as a local control on SNA concentrations since they are directly connected to oxidation capacity and also are involved in alternative loss pathways for nitrate radicals (Aksoyoglu et al., 2017; Womack et al., 2019). Therefore, nitrate formation depends not

only on the amount of $NO_x$ emitted but also on the amount of ammonia and sulfate, ambient conditions (RH and temperature), and VOC and oxidant concentrations. Also relevant are the loss processes, which include dry and wet deposition (affecting both SNA and its precursors) and chemical losses (e.g., uptake by dust, nitrate photolysis). These formation and loss processes, and in turn SNA concentrations, are expected to respond to future changes in precursor emissions and climate (Dawson et al., 2007; Pye et al., 2009; Vasilakos et al., 2018; Aksoyoglu et al., 2020), but

predicting the magnitude and direction of the response depends on how well models capture the complex, non-linear system that describes the lifecycle of atmospheric SNA.

Global atmospheric chemistry models incorporate these mechanisms of SNA production and loss. Past studies have evaluated the SNA simulation in a range of models using surface observations and aircraft campaigns; the results across models can vary substantially, particularly for nitrate (Mezuman et al., 2016; Bian et al., 2017; Chen

et al., 2019; Nault et al., 2021; Reifenberg et al., 2022). Large variations in how nitrate production, partitioning, and loss is described drives differences in simulated nitrate, which can result in modelled total nitrate burden (fine + coarse PM) varying by a factor of 13 (Bian et al., 2017), with some models underestimating nitrate and others overestimating nitrate. We also note that many global models neglect the formation of ammonium nitrate entirely (Gliß et al., 2021; Thornhill et al., 2021). Generally, the sulfate simulation is more consistent and reliable across the different models

(Bian et al., 2017; Nault et al., 2021).

In this study, we use a single model (GEOS-Chem) to systemically evaluate SNA performance. Previous assessments of the global chemical transport model GEOS-Chem have focused on one region or used one specific field campaign. These model evaluation studies have found sulfate is well-captured and that ammonium and nitrate are overestimated in Europe (Park et al., 2004), the US (Park et al., 2004; Heald et al., 2012; Zhang et al., 2012), and

over South Korea (Travis et al., 2022; Zhai et al., 2023). More localized analyses in the US have shown exceptions to this trend, with underestimates in simulated nitrate in California (Heald et al., 2012; Schiferl et al., 2014) and an unbiased nitrate simulation in the Northeastern US in wintertime (Shah et al., 2018). Various theories have been suggested to explain these model biases, including: deficient emissions inventories (Park et al., 2004; Schiferl et al., 2014), underestimated deposition of $HNO_3$ (Heald et al., 2012; Travis et al., 2022), overestimated $N_2O_5$ hydrolysis

(Zhang et al., 2012; Heald et al., 2012), and uptake of acidic gases on coarse dust (Heald et al., 2012; Zhai et al., 2023). These studies provide insight into some of the key processes that may be misrepresented or missing from models such as GEOS-Chem which are adversely affecting simulated SNA concentrations. However, their local focus with various model versions (including changing descriptions of the chemistry and meteorology), make it challenging to generalize these results. Here, we use a suite of 11 aircraft campaigns spanning multiple regions of the world to

provide a more comprehensive and consistent global evaluation of GEOS-Chem SNA performance. We also explore the key processes controlling SNA concentrations, identifying those that may contribute to model bias.

## 2   Description of Observations

This study explores observations from 11 airborne campaigns that span different regions of the world and almost two decades (2004–2019). As a result, these campaigns represent a wide range of chemical regimes and
emissions scenarios. The campaigns are listed in Table 1, including the dates, locations, and primary references. These campaigns were selected because they all 1) share a common measurement technique for SNA concentrations and 2) are not representative of remote conditions, and thus generally have higher concentrations of SNA that are well above detection limits. The campaigns all took place in the Northern Hemisphere in one of three general regions: North America (NA), Europe (EU), or Asia (AS). There are at least two campaigns in each area, but with a large geographical
sampling bias (>50% of the campaigns) towards campaigns in the NA region, particularly over the US. Figure 1 shows the campaign flight tracks. Also in Fig. 1 are panels for each campaign with a pie chart representing the fractional contribution of all three SNA species to the total measured SNA (measurements described below). Below each pie chart is the mean observed total SNA. Units are reported in $\mu g/sm^3$, standardized at STP (P = 1013.25 hPa; T = 273.15 K). To make a more direct comparison across campaigns with varying aircraft ceilings, only points below 5 km are
included in Fig 1. The total SNA concentrations are highest for KORUS-AQ, EUCAARI, MILAGRO, ADRIEX, and SENEX, indicative of the more significant influence of anthropogenic outflow during these campaigns. Generally, sulfate is the largest contributor to total SNA across all 11 campaigns. The nitrate fraction is higher for the three campaigns with the highest SNA concentrations (KORUS-AQ, EUCARRI, and MILAGRO), as well as for CalNex (associated with higher agricultural emissions) and WINTER (associated with colder temperature favouring particle
phase nitrate).

While we focus on campaigns influenced by anthropogenic sources, biomass burning also impacted some of the campaigns (i.e., FIREX-AQ, DC3, and MILAGRO). For FIREX-AQ, the main objective was to improve understanding of the impact of fires on air quality and climate, so both wildfires and prescribed agricultural burning in the US were intentionally sampled. The EMeRGe-EU and EMeRGe-AS campaigns were explicitly interested in air
quality downwind of megacities in Western Europe and Southeast Asia, respectively. We do not include the transit flights for the EMeRGe-AS campaign (corresponding to the flights on the first and last days between Germany and the United Arab Emirates). Other transit flights during EMeRGe-AS between megacity centers in Southeast Asia are included, which involved the sampling of cleaner, ocean air. Similarly, some flights for WINTER measured cleaner air over the Atlantic Ocean.
Across all the campaigns, Aerodyne aerosol mass spectrometers (AMS; Canagaratna et al., 2007) measured sulfate, nitrate, and ammonium concentrations. An AMS measures sub-micron, non-refractory particles with approximately 30% uncertainty for SNA species (Bahreini et al., 2009). Use of a single measurement technique is expected to reduce potential measurement bias between campaigns, though differences in instrument operation and

models (Q-AMS, C-ToF-AMS, HR-ToF-AMS; see Supplementary Table 1 for AMS used in each campaign) may generate some variation.

The nitrate concentrations from the AMS include inorganic and organic nitrate; we use total nitrate in our analysis since the split between inorganic and organic nitrate is not available for all the campaigns. Previous work has shown that the percentage of total nitrate that is organic is highly dependent on total nitrate concentrations, ranging from 0% at highest urban influence to 100% at the cleanest conditions (Day et al., 2022). Given our selection of campaigns that are anthropogenically influenced, we expect inorganic nitrate to dominate total nitrate. We comment further on this in Sect. 3.1 and 4. Similarly, small fractions of the AMS sulfate may be due to organosulfates (Schueneman et al., 2021), and very small fractions of the AMS ammonium may be due to amines (Ge et al., 2023), but these apportionments are not typically reported and possible contributions are neglected here.

We retain only the data points that have valid measurements for sulfate, nitrate, and ammonium. Observational data is filtered to remove plumes (sulfate, nitrate, ammonium concentrations > their respective 95th percentile) since the model is unable to capture these sub-grid processes successfully (Rastigejev et al., 2010). Observations are then averaged from their original resolution (using 1 minute merge files when they are available) to the temporal and spatial resolution of the model. The majority of sampling occurred during the day, but some campaigns had more nighttime flights (e.g., 55% of the valid points for WINTER are at nighttime). After filtering and averaging, there remain 22,616 unique data points that are used in our model-observation comparison.

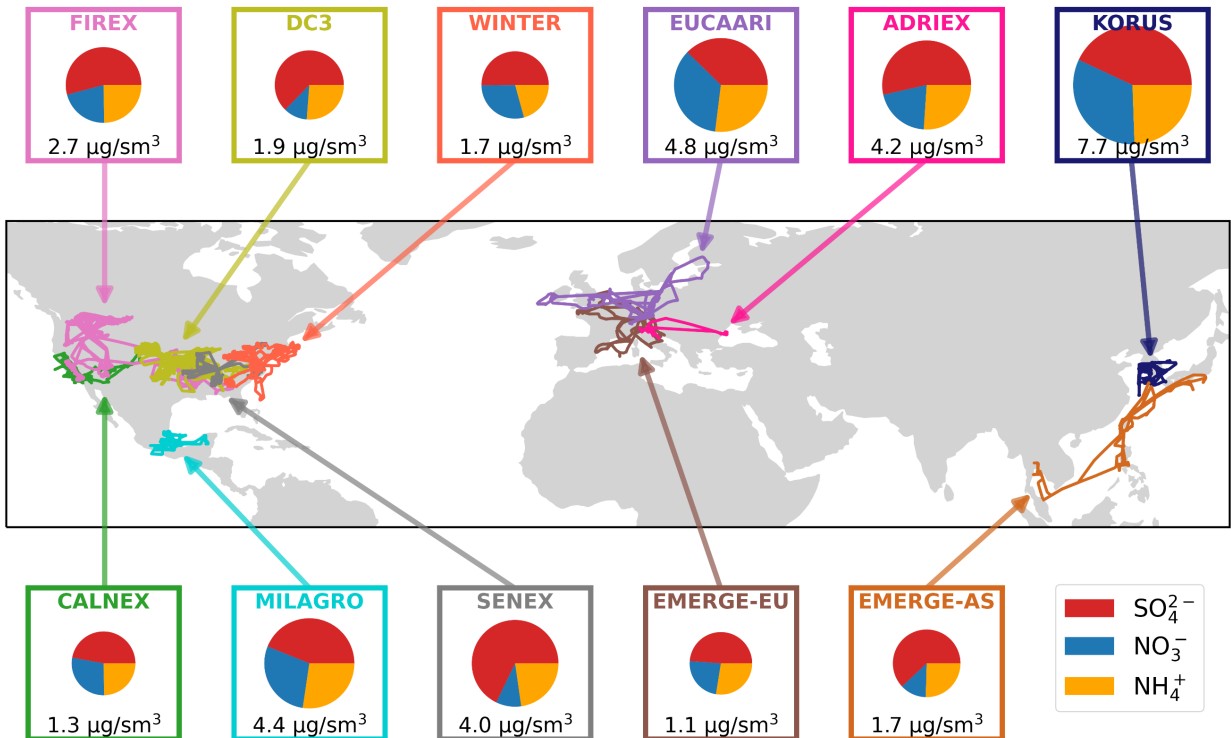

**Figure 1.** *Flight tracks for the airborne campaigns used in this analysis. Pie charts show mean relative contributions of sulfate (red), nitrate (blue), and ammonium (yellow) to total SNA for each individual campaign. The area of the pie charts are scaled based on the mean total SNA for each campaign, which is also reported below the pie chart. Only points below 5km are included. Information about the year and season for each campaign are included in Table 1.*

*Table 1. Details of all the campaigns used in this study, including the dates, regions, and primary references.*

| Campaign | Dates (Season) | Region | Ref. |
|---|---|---|---|
| Aerosol Direct Radiative Impact Experiment (ADRIEX) | 27 Aug – 6 Sept 2004 (fall) | Italy & S Europe | Crosier et al. (2007) |
| Megacity Initiative: Local and Global Research Observations (MILAGRO) | 4 – 31 Mar 2006 (spring) | Mexico City | DeCarlo et al. (2008) |
| European Integrated Project on Aerosol Cloud Climate and Air Quality Interactions (EUCAARI) | 6 – 22 May 2008 (spring) | N & NW Europe | Morgan et al. (2010) |
| California Research at the Nexus of Air Quality and Climate Change (CalNex) | 30 Apr – 22 June 2010 (spring and summer) | California, US | Ryerson et al. (2013) |
| Deep convective clouds and chemistry (DC3) | 18 May – 22 June 2012 (spring and summer) | SE US | Barth et al. (2015) |
| Southeast Nexus-Studying the Interactions between Natural and Anthropogenic Emissions at the Nexus of Climate Change and Air Quality (SENEX) | 26 June – 10 July 2013 (summer)[1] | SE US | Warneke et al. (2016) |
| Wintertime INvestigation of Transport, Emissions, and Reactivity (WINTER) | 3 Feb – 13 Mar 2015 (winter and spring) | NE US | Schroder et al. (2018) |
| Korea–United States Air Quality (KORUS-AQ) | 1 May – 10 June 2016 (spring and summer) | South Korea | Nault et al. (2018) |
| Effect of Megacities on the transport and transformation of pollutants on the Regional to Global scales in Europe (EMeRGe-EU) | 11 – 28 July 2017 (summer) | S & Central Europe | Andrés Hernández et al. (2022) |
| Effect of Megacities on the transport and transformation of pollutants on the Regional to Global scales in Asia (EMeRGe-AS) | 10 Mar – 9 Apr 2018 (spring) | SE & E Asia | Andrés Hernández et al. (2022) |
| Fire Influence on Regional to Global Environments and Air Quality (FIREX-AQ) | 22 July – 5 Sept 2019 (summer and fall) | W & Central US | Warneke et al. (2023) |

[1] Full campaign ran from 3 June – 10 July 2014, but we remove points before June 26th because sensitivity issues with the AMS caused ammonium to be systematically higher than other species for earlier flights (Liao et al., 2017).

## 3 Model description

### 3.1 General Description

We use the GEOS-Chem chemical transport model version 13.3.4 (DOI: 10.5281/zenodo.5764874). Full-year simulations are performed at $2° \times 2.5°$ horizontal resolution while the campaign simulations make use of a finer resolution of $0.5° \times 0.625°$ nested grid driven by boundary conditions from global $2° \times 2.5°$ simulations. The model vertical domain is resolved into 47 hybrid-sigma layers extending from the surface to approximately 80 km altitude. All of the simulations are driven by the MERRA-2 assimilated meteorological data product from the NASA Goddard Global Modeling and Assimilation Office (GMAO). Boundary layer mixing is described using a non-local mixing scheme (Lin and McElroy, 2010). Following recommendations from Philip et al. (2016), timesteps are 20 minutes for chemistry and 10 minutes for transport for the global simulations and 10 minutes for chemistry and 5 minutes for transport for the nested simulations.

GEOS-Chem includes a detailed gas-phase chemistry coupled with the sulfate-nitrate-ammonium aerosol system (Park et al., 2004; Pye et al., 2009), with updates to $HO_2$ uptake (Mao et al., 2013) and the reactive uptake of $NO_2$, $NO_3$, and $N_2O_5$ by aerosols and clouds (Holmes et al., 2019; McDuffie et al., 2018). Dust and sea salt aerosols are separated into different size bins (4 bins for dust: 0.1–1.0 μm, 1.0–1.8 μm, 1.8–3.0 μm, 3.0–6.0 μm; 2 bins for sea salt: 0.01 – 0.5 μm, 0.5 – 8 μm). Sodium is calculated as a fraction of fine sea salt aerosol in GEOS-Chem (39.7% by weight of sea salt). The model uses a bulk aerosol scheme with fixed log-normal modes to describe the size distribution of aerosols (Martin et al., 2003). A resistors-in-series scheme is used to describe gas dry deposition (Wesely, 1989; Wang et al., 1998) and size-dependent aerosol dry deposition (Zhang et al., 2001; Emerson et al., 2020). The wet deposition scheme includes rainout, washout, and scavenging in moist convective updrafts for aerosols and gases (Amos et al., 2012; Liu et al., 2001; Wang et al., 2011, 2014). Thermodynamic partitioning between the gas and particle phase is described by the thermodynamic equilibrium aerosol model ISORROPIA II (Fountoukis and Nenes, 2007; Pye et al., 2009). ISORROPIA II is run using the default, metastable mode, which assumes that all inorganic salts exist on the upper branch of the hygroscopic hysteresis curve. Acid uptake on dust (Fairlie et al., 2010) and nitrate photolysis (Shah et al., 2023) are optional processes in GEOS-Chem version 13.3.4 which we do not include in our model evaluation; however, we explore the effect of both of these processes on SNA in Section 5.5. When examining the impact of acid update on dust, we include nitrate and sulfate on dust in the smallest size bin (≤1 μm) in our model-observation comparisons.

To match the observations, organic nitrate from the model (from isoprene and monoterpene precursors) is also included in nitrate. We use the complex scheme for organic aerosols described by Pai et al. (2020). However, we note that for the campaigns in this work, organic nitrate is a minor constituent of simulated total nitrate (median organic nitrate contribution is 0.1% of total nitrate). The largest median organic nitrate fraction is simulated during SENEX (7.4% of total nitrate), which was heavily influenced by biogenic sources in the Southeast US.

Each GEOS-Chem simulation is matched to the specific time and location of each airborne campaign. The majority of the emissions inventories used in this work are year specific. This includes the global anthropogenic emissions (comprising fossil fuel and agricultural sources) from the Community Emissions Database System (CEDS)

v2 inventory, which also provides ship emissions (year-specific up to 2017; Hoesly et al., 2018), biomass burning emissions from GFED4s (van der Werf et al., 2017), volcanic $SO_2$ emissions (Carn et al., 2017), lightning emissions (Murray et al., 2012), sea salt emissions (Jaeglé et al., 2011), offline dust emissions (Meng et al., 2021), and offline soil $NO_x$ emissions (Hudman et al., 2012). Also included are DMS emissions (Lana et al., 2011; Breider et al., 2017), aircraft emissions from AEIC 2009 (Stettler et al., 2011), and natural (soil, ocean, vegetation, wild animals) emissions of $NH_3$ from GEIA (Bouwman et al., 1997). Anthropogenic emissions for the United States are superseded by the EPA's National Emissions Inventory for 2016 (NEI 2016; Henderson and Freese, 2021). These emissions are also year-specific for all our campaign runs, which are based on annual scale factors derived from emissions trends from 2002–2020. By default, the NEI 2016 emissions inventory has weekday/weekend scale factors applied to the $NO_x$ and $SO_x$ emissions. Time-of-day scaling factors are applied to all anthropogenic $NO_x$ and other fossil-burning emissions globally.

### 3.2 SNA Budget in GEOS-Chem

Figure 2 shows the average global simulated distribution of sulfate, total (organic + inorganic) nitrate, and ammonium at the surface and in the mid-troposphere for the year 2018. Only fine sulfate and nitrate (not associated with sea salt or dust) are included to correspond to the fine mode sampling by the AMS. Concentrations peak at the surface for all SNA species over India, East Asia, and Europe (annual mean concentrations $> 8 \mu g/sm^3$), corresponding to regions with large anthropogenic precursor emissions. Smaller enhancements are visible over the US associated with lower emissions (e.g., stricter regulation; Leibensperger et al., 2012). Other identifiable sources include biomass burning, volcanic emissions, and ocean sources (for sulfate). At the surface, SNA dominates (>50%) simulated $PM_{2.5}$ concentrations across large swaths of the globe (Fig. 3), including near large population centers in the Eastern US, Europe, and Eastern Asia. Surface $PM_{2.5}$ has been evaluated in GEOS-Chem previously and it is generally within 50% of the observations (Lee et al., 2017; Weagle et al., 2018; Zhai et al., 2021b). At the 600mb level (Fig. 2), the same regions stand out as at the surface, but concentrations are generally low ($\sim 1 \mu g/sm^3$). In the mid-troposphere, $SO_4^{2-}$ concentrations are higher and more uniform than $NH_4^+$ and $NO_3^-$ reflecting the significant contributions of ocean sources to background $SO_4^{2-}$ and thermodynamics of ammonium and nitrate aerosols compared with sulfate aerosols.

Table 2 summarizes the budget for SNA and their precursors based on a 2018 simulation. All species have a similar lifetime of around 4–5 days. A significant amount of the emitted $SO_x$ (58 TgS/yr) and DMS (19 TgS/yr) is converted to sulfate and then lost to wet deposition (36 TgS/yr). The precursor emissions for $NO_3^-$ and $NH_4^+$ are 50 TgN/yr for $NO_x$ and 68 TgN/yr for $NH_3$. The budgets for sulfate, nitrate, and ammonium are generally within the range reported by Bian et al. (2017). The notable exceptions are that dry deposition of sulfate is lower in GEOS-Chem compared to all the other reported models (2.5 – 7.3 TgS) and that ammonia emissions exceed the range reported for the AeroCom III models in 2008 (47 – 58 TgN/yr) (Bian et al., 2017). Dry deposition of ammonium (see Table 2) is also at the low end of the range reported in Bian et al. (2017) (1.3 – 16.3 TgN). However, across these models (and GEOS-Chem) dry deposition loss generally makes up less than 20% of the total loss due to deposition (Bian et al., 2017). In comparison, dry deposition of the precursor species (i.e., $SO_2$, $HNO_3$, $NH_3$) is more important, contributing >50% of the total deposition loss of these precursors in GEOS-Chem. Other studies have shown that changes to the

dry deposition of these precursors impacts SNA concentrations (Travis et al., 2022); this is discussed further in Section 245 5.3.

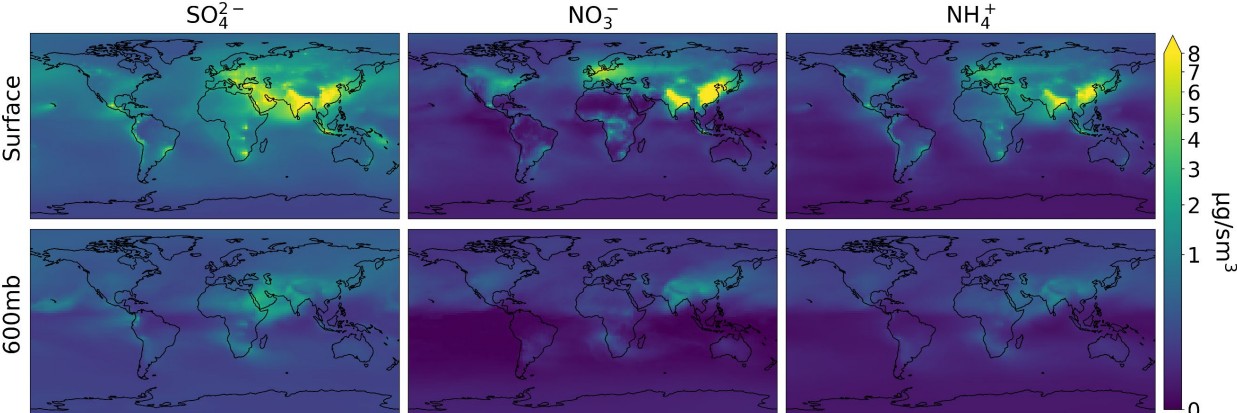

**Figure 2.** *Average annual concentrations of sulfate, nitrate, and ammonium at the surface and in the mid-troposphere (600mb) for 2018.*

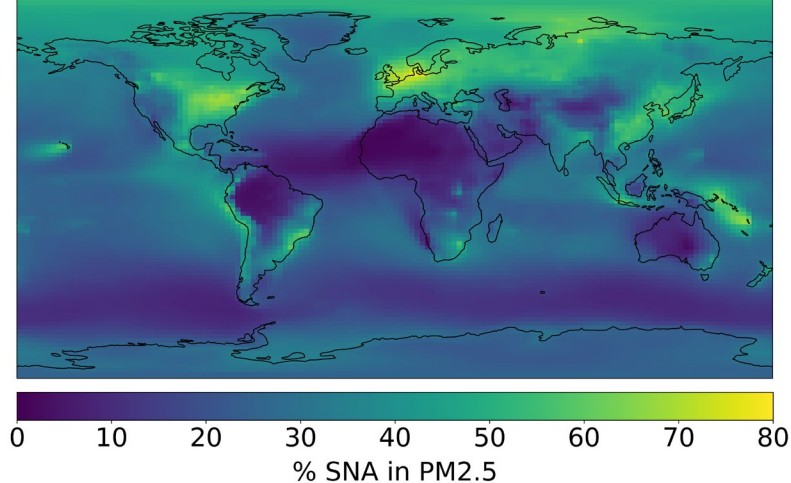

**Figure 3.** *Percent contribution of SNA to annual mean surface PM$_{2.5}$ concentration based on a global simulation for 2018.*

**Table 2.** *Summary of the 2018 global, tropospheric budget in GEOS-Chem for SNA and their precursors. Note NO$_3^-$ corresponds to fine, inorganic + organic nitrate. The lifetime is to dry and wet deposition only.*

|                          | SO$_2$ | SO$_4^{2-}$ | HNO$_3$ | NO$_3^-$ | NH$_3$ | NH$_4^+$ |
|--------------------------|--------|-------------|---------|----------|--------|----------|
| Burden (TgS or TgN)      | 0.3    | 0.4         | 0.3     | 0.09     | 0.2    | 0.3      |
| Wet Dep (TgS/yr or TgN/yr) | 10.2 | 36.4        | 15.8    | 5.9      | 18.0   | 23.3     |
| Dry Dep (TgS/yr or TgN/yr) | 24.2 | 2.1         | 16.7    | 0.7      | 23.9   | 2.2      |

| Lifetime (days) | 3.1 | 4.1 | 3.1 | 5.1 | 1.7 | 4.6 |
|---|---|---|---|---|---|---|

## 4 Model Evaluation

We summarize the model evaluation of inorganic aerosol using two different statistical metrics: the coefficient of determination ($R^2$) and the normalized mean bias (NMB). The ability of the model to capture variability is indicated by $R^2$. NMB is the sum of the differences between each model and observation data point normalized by the sum of all the observations, where a positive (negative) NMB implies the model is overestimating (underestimating) the observations. It provides an idea of the relative bias irrespective of total concentration, which

varies across these different campaigns. These statistics are calculated for the point-by-point comparison between the observations and model or, only where explicitly mentioned, using the vertical profiles. R values (not presented here) are all positive except for those corresponding to the $NO_3^-$ vertical profiles (discussed in detail below) of two campaigns (CalNex and SENEX), where the model and observations show opposite trends with height. Figure 4 shows the $R^2$ and NMB values for all the campaigns and the three SNA species. $R^2$ values range from 0.01 (very poor) to

0.65 (variability in observations reasonably well captured). For all the campaigns, the model performance is best for sulfate ($R^2 = 0.51$, NMB = 0.11) and notably worst for nitrate ($R^2 = 0.22$, NMB = 1.76). Model performance for ammonium generally lies between that for nitrate and sulfate ($R^2$ and NMB are 0.43 and 0.66 for all campaigns combined), reflecting the strong role that these acidic species play in the amount of ammonium formed. Better performance is expected for sulfate because the formation rates (under typical atmospheric conditions) are well-

understood and concentrations are not controlled by variable gas-particle partitioning. Figure 4 also demonstrates spatial variation in performance, with consistent high biases across all three species for the campaigns in Asia and Europe. In contrast, there is more variability by campaign and by species for the North American campaigns, with no apparent relationship in bias for these campaigns with year, season, or source influence. However, the high nitrate bias is more consistent with extreme overestimates (NMB > 2) seen across all three regions. When nitrate is scaled

down based on the NMB across all the campaigns (NMB = 1.76), average $PM_{2.5}$ concentrations across Northern Hemisphere land decrease by 3.4%, with maximum reductions of 25% in Eastern US and East Asia and 33% in Europe (Fig. S1 in the Supplement).

      We examine if there is a connection between nitrate bias and the model gas ratio (Fig. S2), which is the ratio of free ammonia ($[NH_x]-2[SO_4^{2-}]$) to total gas + particle nitrate (Ansari and Pandis, 1998). A GR > 1 indicates that the

system is $HNO_3$ limited, 0 < GR < 1 the system is $NH_3$ limited, and GR < 0 the system is extremely $NH_3$ limited and indicates that sulfate is not fully neutralized. When $NH_3$ is extremely limited, $NO_3^-$ concentrations are lower and there is consistent negative bias in the simulated $NO_3^-$. This suggests that GEOS-Chem has an excessively strong $NH_3$ limitation that is inhibiting some nitrate formation in these relatively clean (low SNA concentration) regions. However, these comparisons are also subject to measurement detection limits. The majority of the observations are

characterized by GR > 0, which includes both ammonia limitation (0 < GR < 1) and $HNO_3$ limitation (GR > 1); the simulated nitrate is positively biased in both cases, which indicates that the model bias is not the result of one specific precursor limitation.

Figures 5 and 6 show the vertical profile of median sulfate and nitrate (respectively) for each campaign. For sulfate, there are some modest under- and overestimates in magnitude across the campaigns. However, the model captures the generally consistent sulfate vertical profile shape, with most showing a peak at the surface and decreasing concentrations with altitude. The vertical profile of $SO_2$ (Fig. S3) is also well captured by the model, but there is limited model skill for this species on a point-to-point basis ($R^2 = 0.31$) which may degrade the sulfate simulation. The ratio $SO_4^{2-}/SO_x$ (for campaigns that have $SO_2$ data) is well-captured for 4 of the 9 campaigns, but it is substantially overestimated for the remaining campaigns (CalNex, WINTER, MILAGRO, EMeRGe-EU, and EMeRGe-EU), particularly above the boundary layer (Fig. S4). For CalNex and MILAGRO, $SO_2$ is underestimated and $SO_4^{2-}$ is overestimated (while total $SOx$ is well-captured), suggesting that oxidation may be overly rapid; for the other campaigns there is no evident relationship in the bias.

The shape of the observed vertical profile is less consistent for nitrate. For most campaigns, the model generally captures the vertical profile, albeit often with high biases both near the surface and aloft, especially for the European and Asian campaigns. However, in the case of the CalNex and DC3 campaigns, the model predicts peak nitrate concentrations aloft, which is not seen in the observations. The simulated nitrate also shows higher variability (larger IQR) compared to the observations and modelled sulfate. As indicated by the $NO_3^-$ vertical profile for CalNex, this campaign measured many negative $NO_3^-$ concentrations (25% of all points), especially at higher altitudes (greater than 3km all altitude bins have > 60% negative points). While we do not remove these points for any of our model-observation comparisons, we note that the bias would remain but be modestly decreased if points below the detection limit were removed from our analysis. Observed and modelled ammonium profiles (Fig. S7) exhibit similar trends to nitrate including the high-altitude peaks in simulated nitrate seen for CalNex and DC3, but generally exhibit less bias than nitrate.

The campaigns are influenced by a range of conditions which dictate the relative importance of particular processes. For example, some campaigns like EUCAARI and ADRIEX had strong inversions at the top of the BL which led to increasing concentration of nitrate with height within the BL. Restricting the focus to points above the model-defined planetary boundary layer height (71% of points) shows an improvement in $R^2$ for $NO_3^-$ across all campaigns (increases by <0.01 to 0.13 relative to when all points are used), which implies that there is more model skill at capturing $NO_3^-$ aloft. However, there is also an increase in the bias (NMB for $NO_3^-$ increases to 2.91 across all the campaigns). Some campaigns (e.g., ADRIEX and EUCAARI) are less likely to be influenced by any deficiencies in the description of wet deposition in GEOS-Chem due to the lack of rainfall during the campaign (Crosier et al., 2007; Morgan et al., 2010). Others (e.g. DC3 and FIREX-AQ) may have biases associated with the challenges in capturing convective events. The exaggerated peak in simulated nitrate for DC3 could be associated with missing deposition because the storms are small compared to the spatial resolution of the model (Li et al., 2018). Consistent biases in vertical transport or precipitation are unlikely to explain the nitrate bias across these campaigns given that the model reproduces the expected vertical profiles for soluble species such as sulfate (Fig. 5) and for insoluble species such as CO (Fig. S8). In what follows, we use the merged dataset to focus on the universal response to processes, however, it is important to acknowledge that local biases in emissions and meteorology may degrade the model performance for individual campaigns, as explored in greater detail in campaign-specific studies.

As described in Sects. 2 and 3.1, the model and observed values for nitrate also include organic nitrate. Median observed nitrate concentrations are generally mid-range (0.05 – 0.7 µg/sm$^3$) across most campaigns and at all altitudes, which implies these are, generally, environments where the relative contribution of organic nitrate could be significant (~20–80%) (Day et al., 2022). However, we find that the model organic nitrate contributes very little to total simulated nitrate concentrations across almost all the campaigns (Sect. 3.1). While this suggests that

improvements to the organic nitrate description in GEOS-Chem are needed (Pai et al., 2020), it also indicates that the large positive bias in simulated nitrate is indicative of even greater deficiencies in the description of inorganic nitrate in GEOS-Chem. Furthermore, measurements of nitrate might be biased high for campaigns that used a C-ToF-AMS (CalNex, EMeRGe-AS, EMeRGe-EU, EUCAARI, SENEX) where the bias in observational nitrate is proportional to the organic mass concentrations (e.g., corrected nitrate measurements were 30% lower than the measured values due

to organics for one SENEX flight; Fry et al., 2018). Correcting for any overestimates in observed nitrate for these campaigns would worsen the model bias in nitrate.

    In what follows, we examine potential causes of SNA model bias, with a focus on the nitrate bias, specifically the role that deposition, thermodynamic partitioning, chemistry, and/or emissions biases may play.

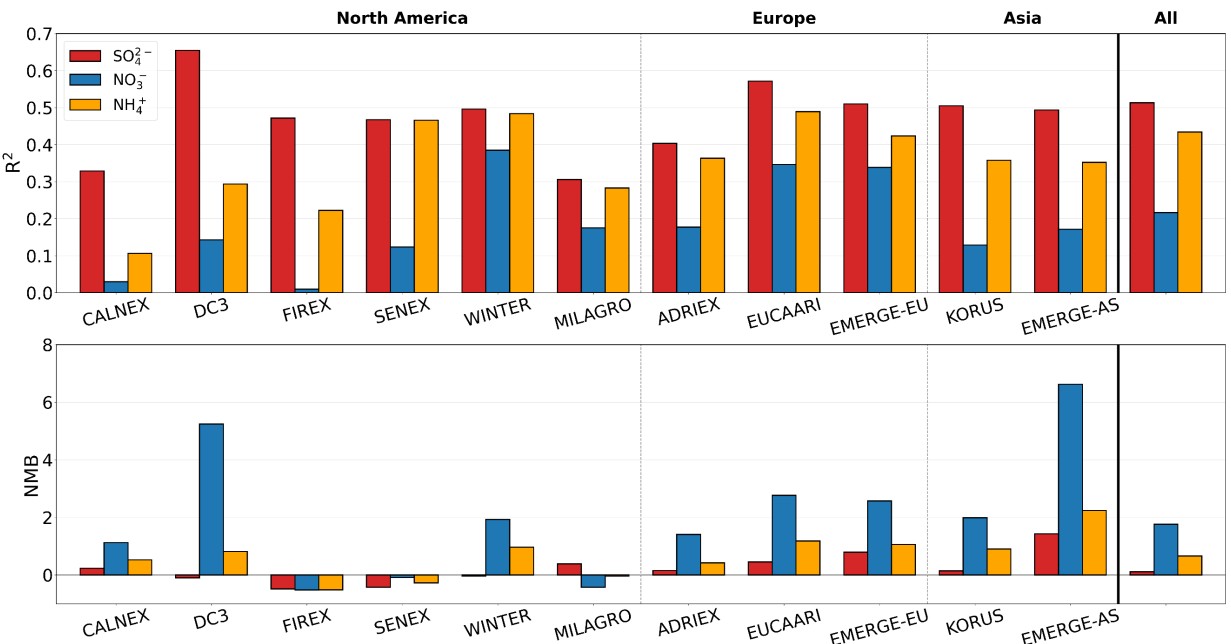

***Figure 4.*** *GEOS-Chem model performance evaluated against each airborne campaign for sulfate (red), nitrate (blue), and ammonium (yellow) reported as R$^2$ and NMB. Campaigns are grouped by the three general regions examined in this study. Model performance for all the campaigns merged into one dataset is shown under 'All'.*

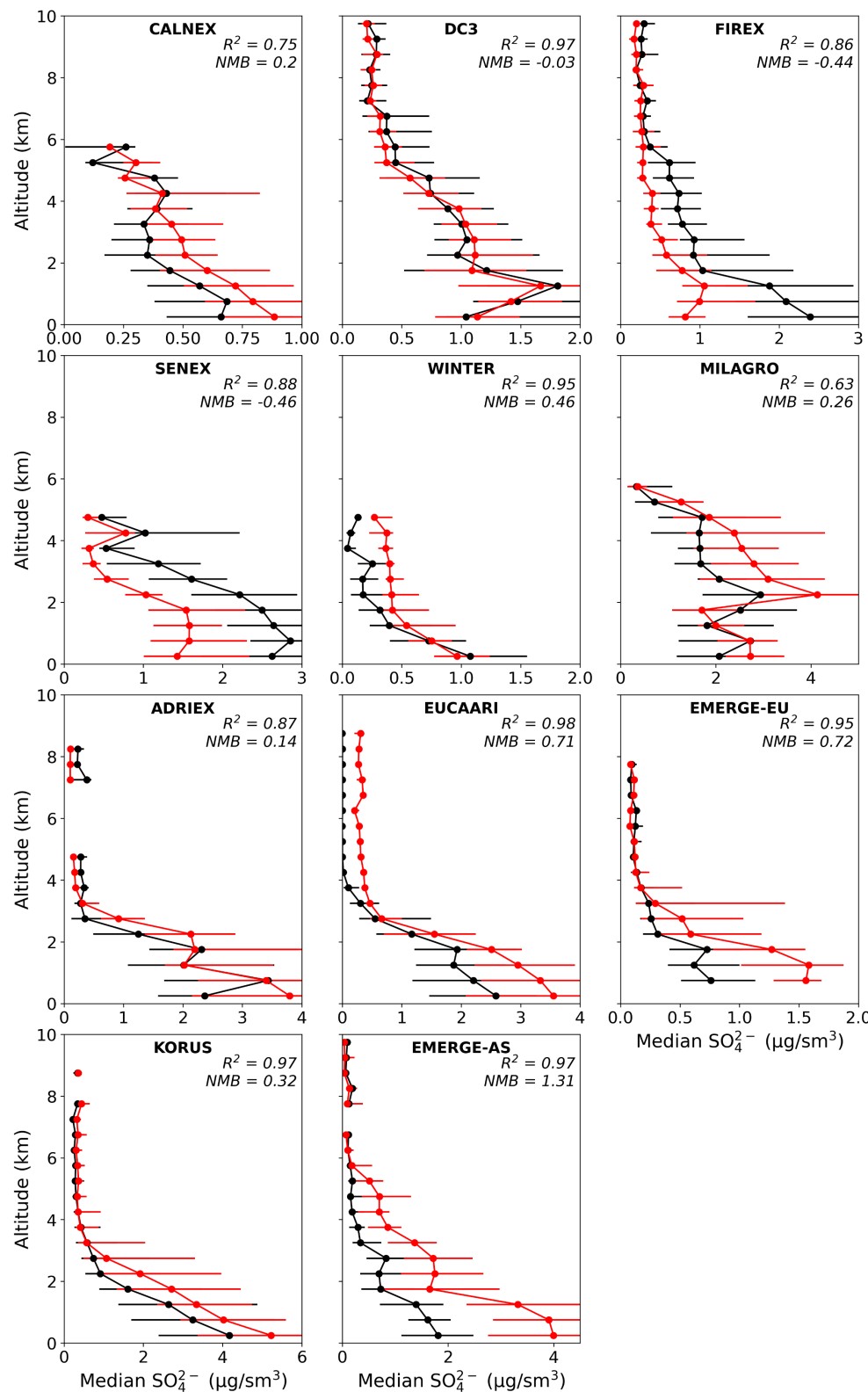

**Figure 5.** *Median vertical profile of observed (black) and simulated (red) sulfate concentrations. Points are binned to the nearest 0.5 km. Error bars represent the interquartile range (IQR). Altitude bins with less than 10 points per bin are not shown. $R^2$ and NMB for the vertical variability is also reported for each campaign.*

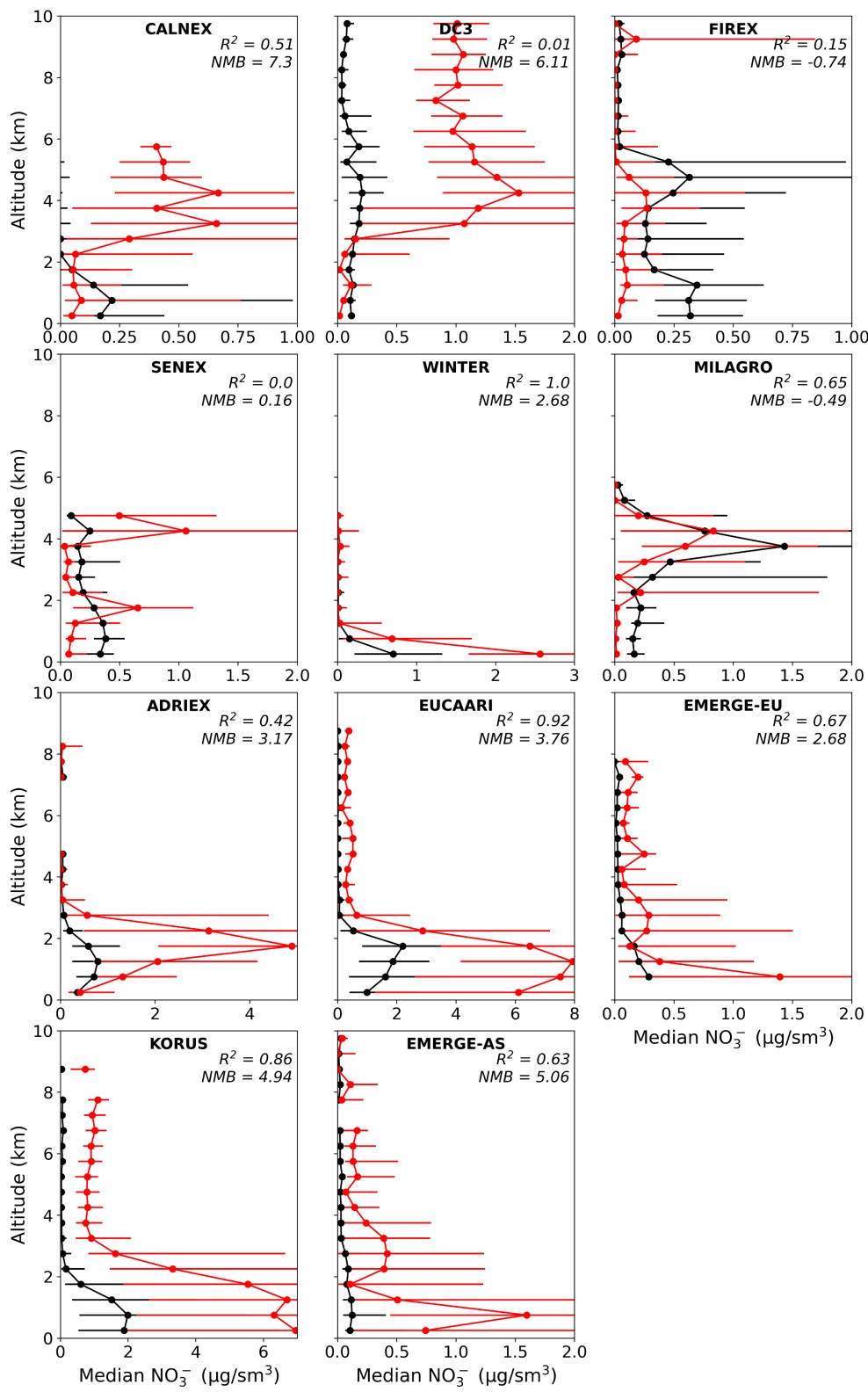

***Figure 6.*** *Median vertical profile of observed (black) and simulated (red) nitrate concentrations. Points are binned to the nearest 0.5 km. Error bars represent the interquartile range (IQR). Altitude bins with less than 10 points per bin are not shown. $R^2$ and NMB for the vertical variability is also reported for each campaign.*

## 5 Investigating Model Bias

### 5.1 Evaluating Thermodynamic Partitioning

#### 5.1.1 Evaluating Thermodynamic Partitioning in ISORROPIA II

First, we examine whether errors in the thermodynamic partitioning, represented via the ISORROPIA II scheme could contribute to some of the model bias. Issues with partitioning, which can also act as a strong control on dry deposition and lifetime of total (gas- + particle-phase) nitrate ($TNO_3^-$) and ammonium ($NH_x = NH_3 + NH_4^+$; Nenes et al., 2021), could contribute to the model SNA bias. ISORROPIA II, as implemented in GEOS-Chem and in forward mode, partitions $TNO_3^-$, ($NH_x$), and chloride ($TCl^- = HCl + Cl^-$), based on the total concentrations of these species, temperature (T), relatively humidity (RH), and sodium and sulfate concentrations. It does not include cations associated with mineral dust ($K^+$, $Ca^{2+}$, and $Mg^{2+}$), which are included in other implementations of ISORROPIA II.

The ability of ISORROPIA II to partition successfully can be evaluated by providing the observations as an input to a standalone version of ISORROPIA II (in forward mode) and comparing the predicted partitioning to the expected partitioning (i.e., the observations). However, none of the campaigns explored here included a complete set of measurements for the relevant species to fully evaluate partitioning. In particular, $NH_3$, HCl, and $Na^+$ were only measured for 2, 3, and 4 of the campaigns, respectively. We do not use the $NH_3$ data collected for WINTER due to issues with the sample collection, as discussed in Guo et al. (2016), nor the $NH_3$ data collected for FIREX-AQ because it only reports enhancements in plumes which are not captured well by the model. Therefore, we undertake our evaluation of partitioning by substituting GEOS-Chem simulated values for these three species for all campaigns. In addition, we only consider the subset of campaigns where $HNO_3$ and $Cl^-$ are measured, which leaves 7 campaigns for our evaluation of ISORROPIA II. We filter the data as described in Sect. 2 and remove any points with missing or negative SNA, T, RH, $HNO_3$, or $Cl^-$ to use as an input to ISORROPIA II. The resulting ISORROPIA II predicted nitrate and ammonium concentrations do not agree perfectly with observations, though the overall NMB is small (Fig. 7). There are three input factors that may contribute to the imperfect performance in Figure 7: the meteorology, the substituted model values, and measurement uncertainties.

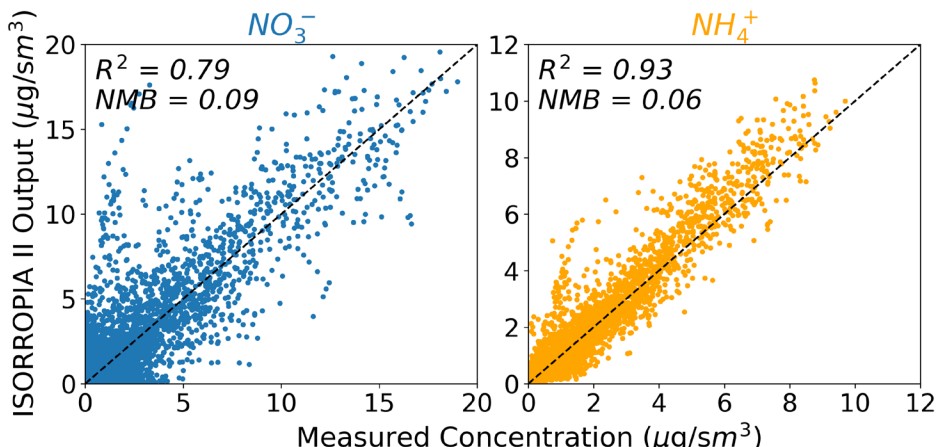

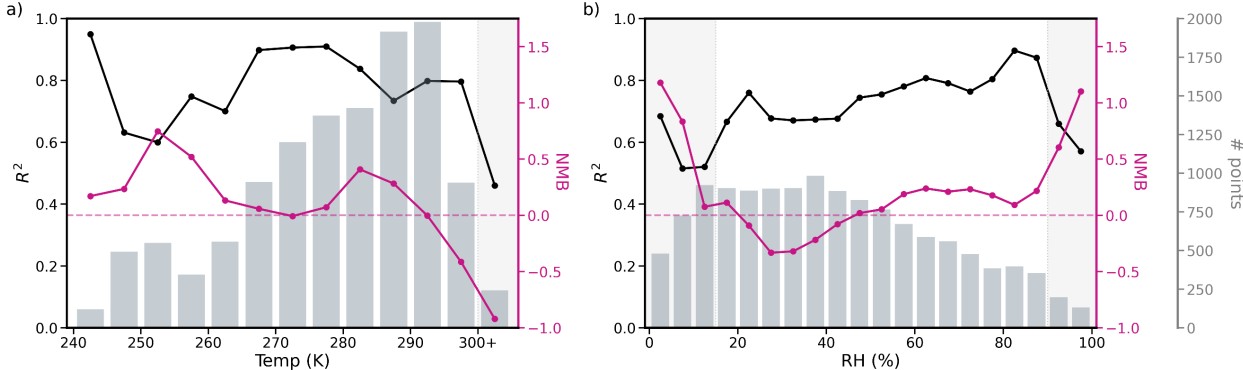

**Figure 7.** *Comparison of the expected (x-axis) to ISORROPIA II-predicted (y-axis) aerosol concentrations. Observations of T, RH, $SO_4^{2-}$, $NO_3^-$, $NH_4^+$, $HNO_3$, and $Cl^-$ are used as inputs into ISORROPIA II. Only the campaigns that include these measurements are represented.*

**Figure 8.** *$R^2$ (black) and NMB (pink) for ISORROPIA II predicted nitrate concentrations (with observations given as input) versus a) temperature and b) RH. Points are binned to the nearest 5K and 5% for the temperature and RH plots, respectively. Dark gray bars indicate the number of points in each bin. Light gray sections of the plot show which ranges of temperature and relative humidity result in worsened performance*

Figure 8 shows the relationship between ISORROPIA II performance ($R^2$ and NMB) and temperature and RH specifically for $NO_3^-$. Performance degrades when RH < 15% or RH > 90% and T > 300 K. Previous work supports these observed limitations of ISORROPIA II's performance at very low humidity where under these conditions the aerosols are less likely to be in a completely liquid state (Ansari and Pandis, 2000; Malm and Day, 2001; Fountoukis and Nenes, 2007; Bertram et al., 2011). Also at very high humidity, there is exponential growth in the particle liquid water, which can lead to large uncertainties in the pH (Malm and Day, 2001; Guo et al., 2015). We therefore filter out these points (retaining only points where T < 300 K and $15 \leq RH < 90\%$) in all subsequent analysis; however, we find that doing so only moderately improves the performance ($R^2$ and NMB) of ISORROPIA II exhibited in Fig. 7 (impact on GEOS-Chem performance discussed in Section 5.1.2).

A more critical, but difficult to assess factor is the use of model substituted values for $NH_3$, HCl, and $Na^+$ concentrations. Figure 9 shows that for the limited campaigns where these species are measured, the model does not capture the observed variability (low $R^2$), and in the case of sodium exhibits significant biases. Observations of sodium are limited, and the only available measurements are for bulk aerosol (<4μm), which does not align with the definition of sodium in GEOS-Chem (fraction of fine mode sea salt); these differences in size cut explain at least some of the discrepancy in Fig. 9. Observations for $NH_3$ are only available for two of the campaigns (SENEX and CalNex). The near zero NMB for $NH_3$ in Fig. 9 is driven by large model overestimates for SENEX, with both over and underestimates for CalNex. The variation in model performance could indicate that regional processes (e.g., emissions) dominate ammonia model bias.

For the two campaigns where $NH_3$ measurements are available, we find that using these as inputs to ISORROPIA II, rather than model values, impacts the comparison between predicted and observed nitrate, with particularly large improvements in the $R^2$ for CalNex (Figs. S9 and S10). Similar tests for $Na^+$ and HCl had negligible

impact on bias and $R^2$, despite the clear inability of GEOS-Chem to capture the observed concentrations of these species (Fig. 9). We note that non-volatile cations, which other than $Na^+$ are not accounted for this implementation of ISORROPIA II, have been shown to shift partitioning, producing an average fine nitrate aerosol surface concentration that is 21% higher than in a simulation with chemically inert dust (Karydis et al., 2016). This increase in nitrate is seen despite also introducing a loss pathway for $HNO_3$ that reduces nitrate formation (discussed in more detail in Sect. 5.5).

Figure 7 does not exhibit a systematic low bias in nitrate, suggesting that for the campaigns considered in this study, neglecting non-volatile cations does not produce noticeable partitioning bias.

    Our evaluation of ISORROPIA II in Figure 7 focuses on the aerosol nitrate and ammonium concentrations since these are the target species for our GEOS-Chem model simulation. A more explicit evaluation of the partitioning would explore the performance of $\varepsilon(NO_3^-)$ ($=NO_3^-/TNO_3^-$) and $\varepsilon(NH_4^+)$ ($=NH_4^+/TNH_x$). However, the rarity of

observed $NH_3$ limits the dataset for which the observed partitioning can be fully assessed. For completeness we evaluate $\varepsilon(NO_3^-)$ and $\varepsilon(NH_4^+)$ using model substituted ammonia concentrations as used in Figure 7. The resulting ISORROPIA II-predicted $\varepsilon(NO_3^-)$ demonstrates little skill ($R^2 = 0.25$), whereas $\varepsilon(NH_4^+)$ is better captured ($R^2 = 0.78$; Fig. S11 in Supplement). We identify no consistent relationship between the low $R^2$ and other variables (e.g. other species, pH, concentrations) across the campaigns.

For the two campaigns with $NH_3$ observations, replacing the GEOS-Chem sourced-$NH_3$ values with the observed $NH_3$ improves $R^2$ for $\varepsilon(NO_3^-)$, but at the cost of worsening $R^2$ for $\varepsilon(NH_4^+)$ (Figs. S9 and S10). We also explore the possibility of using estimated $NH_3$ values for all campaigns. Following Guo et al. (2016), we iteratively solve for $NH_3$ by cycling through different input $TNH_x$ values for ISORROPIA II until the expected concentration of $NH_4^+$ is returned (or it fails to reach a solution). Using these new $NH_3$ values improves agreement between observed

and ISORROPIA II-predicted $\varepsilon(NO_3^-)$ ($R^2 = 0.59$). In particular, we note that we get a similar comparison between model and observed $\varepsilon(NO_3^-)$ for WINTER as in Guo et al. (2016) ($R^2 = 0.61$, NMB = -0.41, and performance is best when RH > 50%). However, these estimated $NH_3$ values greatly, and unrealistically, overestimate the observed $NH_3$ from CalNex and SENEX (NMB = 0.48 and 11.39 respectively).

    The limited evaluation of $\varepsilon(NO_3^-)$ and $\varepsilon(NH_4^+)$ possible here suggests that there may be some unresolved

issues with partitioning as represented by ISORROPIA II. We note that our analysis assumes that the measurements are unbiased, there are no missing bases, and that the system is in thermodynamic equilibrium. Representation of non-equilibrium thermodynamics can introduce some improvement in model bias for SNA but can also worsen model performance (Rosanka et al., 2024), suggesting that the missing non-equilibrium process in this work is unlikely a large contributor to the model bias shown here. More work is needed to fully evaluate ISORROPIA II performance

for ammonium nitrate across a range of conditions, including using a full suite of gas and aerosol phase measurements. However, for the purposes of this broader investigation into ammonium nitrate performance within GEOS-Chem, we conclude that partitioning is not a dominant source of bias in the $NO_3^-$ concentration comparisons (Fig. 7) and restricting the RH and T range can improve agreement between observations and model (Fig. 8).

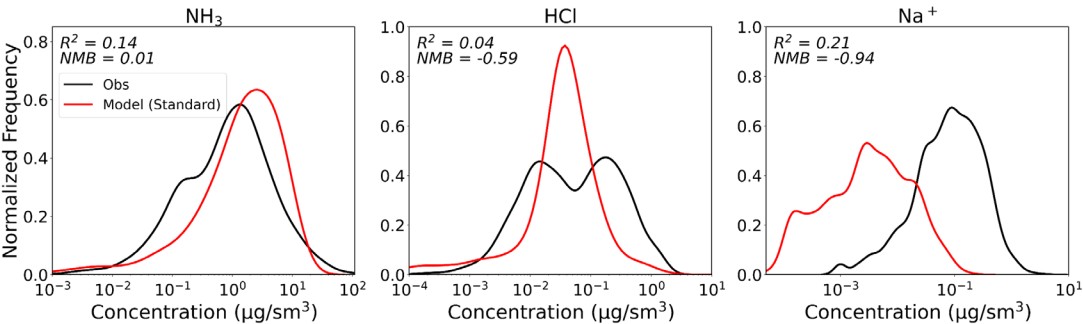

Figure 9. *Distribution of the observed and model values for $NH_3$, $HCl$, and $Na^+$ with reported $R^2$ and NMB.*

### 5.1.2 Evaluating Thermodynamic Partitioning in GEOS-Chem

In addition to the comparisons shown in Fig. 9, here we explore whether there are biases in other model parameters that control thermodynamic partitioning and to what extent this may contribute to the GEOS-Chem biases in nitrate. Figure 10 shows the spread in these ISORROPIA II inputs for both the observations and the model. Where measured, $HNO_3$ is generally overestimated by the model (NMB = 0.44). This could result from overestimated precursor emissions, excessive chemical production, or alternatively, underestimated loss of $HNO_3$ that could generate a high bias in $HNO_3$ and, in turn, $NO_3^-$ (discussed later). We also note that there is no systematic bias in the simulated $NO_3^-$/$TNO_3$ (Fig. S6). The over and underestimates in this ratio are consistent with the $NO_3^-$ bias seen in Figure 6, and thus are not indicative of a partitioning bias, further supporting the analysis of the previous section. The model underestimates $Cl^-$ and does not capture the observed variability (low $R^2$). Temperature is very well captured by the model (high $R^2$, low NMB). The distribution of RH is similar between model and observations in Fig. 10, but the lower $R^2$ value indicates that there are differences in RH on a point-by-point basis. Some of the disagreement between observed and model RH can be explained by the observed RH being defined with respect to water, while the model RH is defined with respect to the relevant phase (ice, water, or a combination of the two) depending on temperature. This leads to greater discrepancies in RH aloft (Fig. S12). However, converting model RH to be with respect to water does not significantly alter ISORROPIA II predicted partitioning and therefore does not contribute to the model bias.

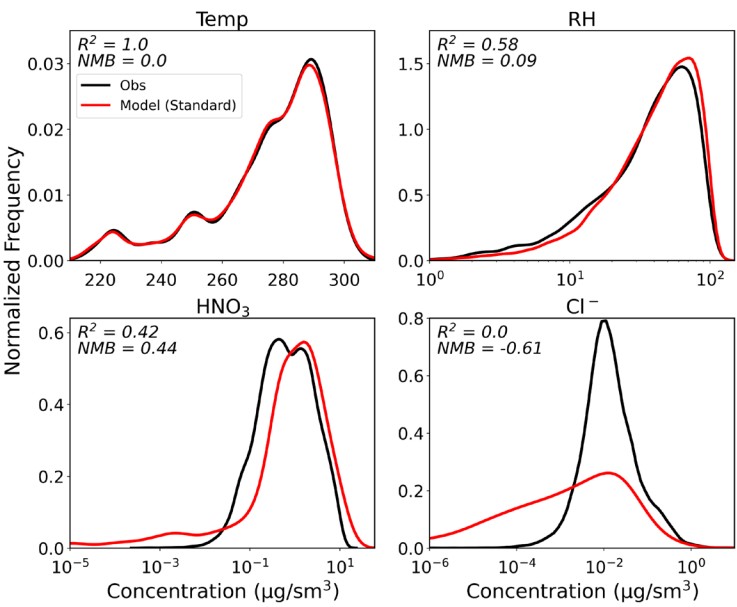

**Figure 10.** *Distribution of the observed and model values for the different variables needed as input to ISORROPIA II with reported $R^2$ and NMB.*

460   As in the previous section, we filter by RH and temperature (retaining points with T < 300K and 15 ≤ RH < 90%) since Fig. 8 confirms that ISORROPIA II may not appropriately capture thermodynamic partitioning at these extremes of the observed T and RH. Twenty percent of the data points are eliminated by this filtering, with most of the points lost (72%) from low altitudes (< 4km). This filtering has a small effect on the model performance shown in Figs. 4–6. Sulfate performance is relatively unchanged (new $R^2$ and NMB of 0.54 and 0.13 for all campaigns 465 combined). $R^2$ for all campaigns combined is decreased very minimally for $NO_3^-$ (0.22 to 0.21) and $NH_4^+$ (<0.01 difference). The largest change after filtering is the reduction in $NO_3^-$ NMB from 1.76 to 1.70. A small fraction of the model high nitrate bias can therefore be explained by the temperature and RH range limitations, specifically for the partitioning by ISORROPIA II in GEOS-Chem. The comparison of ISORROPIA II-predicted pH using the observations and using the model values is also improved after filtering by T and RH ($R^2$ goes from 0.28 to 0.32 and 470 NMB from 0.32 to 0.19, see Fig. S13). For the remainder of this study, we remove points in these temperature and RH extremes and explore what processes might be responsible for the remaining nitrate bias.

   We now test how the model values for T, RH, $HNO_3$, $Cl^-$, $SO_4^{2-}$, $NO_3^-$, and $NH_4^+$ impact the partitioning and contribute to the high $NO_3^-$ bias in GEOS-Chem. Figure 11 shows a series of sensitivity tests where different combinations of modelled and observed values were given as an input to standalone ISORROPIA II. The bias of each 475 sensitivity test, relative to the "true", observed $NO_3^-$ and $NH_4^+$, are represented by the x and y axes respectively.

   The 'Obs' sensitivity case refers to when all the possible observations available for each campaign are used as input to ISORROPIA II. As in the previous section, we only use the campaigns that have $HNO_3$ and $Cl^-$ measurements. We see that the ISORROPIA II-predicted nitrate and ammonium are only slightly high biased compared to observations when ISORROPIA II is driven by the entire (but incomplete) set of observed concentrations 480 and meteorology (also seen in Fig. 7). We attribute this slight bias to the unmeasured species across the dataset in

Sect. 5.1.1. The 'Model' test case refers to using only the output from GEOS-Chem along the flight tracks as input to ISORROPIA II. The model is biased high compared to the observations, consistent with the results of the model evaluation in Sect. 4.

To identify whether any specific parameter drives the model bias, we substitute model values with observed values one at a time. When we replace the model temperature with the observed temperature, as in the 'Obs T' run, we see a negligible impact on the partitioning, as expected given the match in observed and MERRA-2 temperature (Fig. 10). Similarly, substituting the observed $HNO_3$, $Cl^-$, and RH for model values (in three separate tests) produces little change in the thermodynamic partitioning, despite the biases seen between the model and observations in Fig. 10. As expected, the high bias in model $HNO_3$ shifts the partitioning towards more particle phase. Despite an apparent high bias in model RH (NMB = 0.09, see Fig. 10), substituting observed RH for model RH results in less particle phase, which is associated with a low bias in model RH at higher $NO_3^-$ and $NH_4^+$ concentrations.

Using observed sulfate, which is generally lower than the model, as an input to ISORROPIA II produces less ammonium, but more nitrate, as expected. However, the changes are relatively modest and do not suggest that sulfate model biases are responsible for the substantial biases in ammonium nitrate seen in GEOS-Chem. Greater improvements in predicted nitrate and ammonium concentrations result from using the observed ammonium or, more noticeably, the observed nitrate. The least biased ISORROPIA II prediction results from substituting in the observed sulfate, nitrate, and ammonium ('Obs SNA'), which nearly removes all bias for both nitrate and ammonium. This indicates that the GEOS-Chem model bias in nitrate and ammonium is largely a result of model SNA itself, rather than partitioning biases driven by meteorology, other aerosol constituents, or gas-phase precursors. However, we note that without a complete set of observed $NH_3$ measurements, we cannot fully assess how biases in this species and the associated emissions may play a role in this model bias. We also note that while the magnitude of the NMB in $NO_3^-$ and $NH_4^+$ shown in Fig. 11 are sensitive to the subset of campaigns used, the general trends remain the same (i.e., changes in T, RH, $Cl^-$, and $HNO_3$ have low impact, change in SNA has the largest).

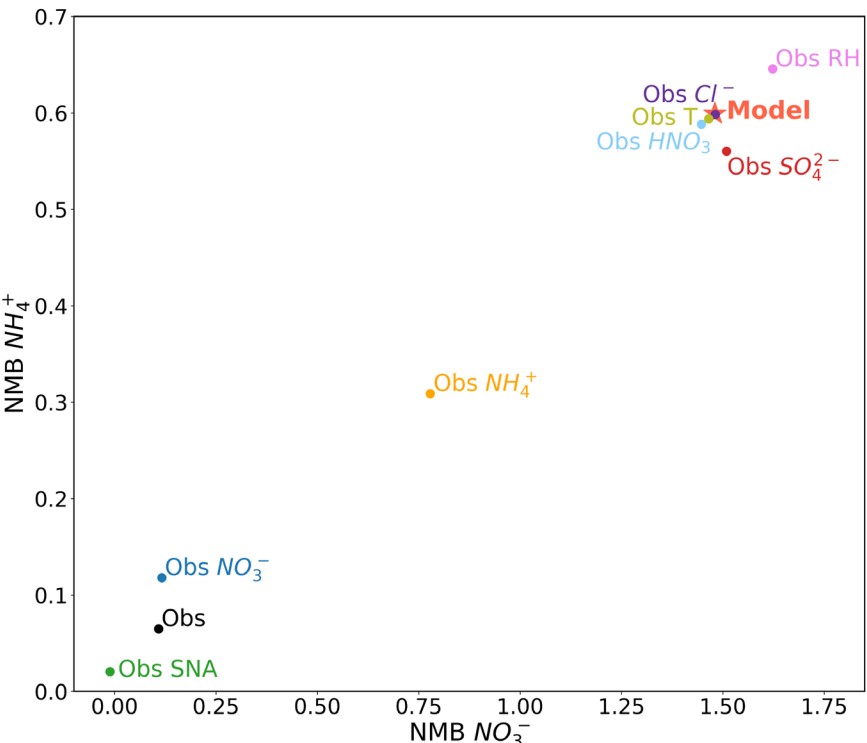

**Figure 11.** *Bias in NO₃⁻ and NH₄⁺ associated with different sensitivity tests with ISORROPIA II using all the available observed values ('Obs'), all modelled values ('Model'), and when different observed values are substituted in for model values. Data is filtered to retain points where model T < 300 K and 15 ≤ model RH < 90%.*

The analyses above suggest that the GEOS-Chem model overestimate of nitrate (and ammonium) is likely the result of an excessive source or an underestimated or missing loss process for nitrate itself. We leverage the fast-run time of standalone ISORROPIA II to run a multitude of sensitivity tests to explore how much $TNO_3^-$ and $NH_x$ would need to change in GEOS-Chem to improve model performance. Figure 12 shows the model performance, using NMB as the metric, for $NH_4^+$ and $NO_3^-$ when the simulated values of $TNO_3^-$ and $NH_x$ are scaled. All campaigns are included and are grouped by region to capture how changes on a regional scale could improve model performance. The NMB for the sum of ammonium and nitrate is also shown, where the swaths of gray (where NMB is near zero) indicate that there are different scalings of $TNO_3^-$ and $NH_x$ that would all result in a similarly "most improved" simulation for both ammonium and nitrate. All three regions exhibit the same pattern, but the scaling factors are shifted up/down depending on regional model biases. For example, the North American campaigns, which are generally less biased (Fig. 4), require the least change (a 25% reduction of $TNO_3^-$ and/or $NH_x$) to eliminate the bias. In contrast, the simulation would be most improved for the European and Asian campaigns with significant cuts (up to 50–75%) to $TNO_3^-$, $NH_x$, or both. In the coming sections, we explore how different production and loss processes in GEOS-Chem could reduce $TNO_3^-$ and $NH_x$ in GEOS-Chem and, in turn, produce an improved simulation for ammonium nitrate.

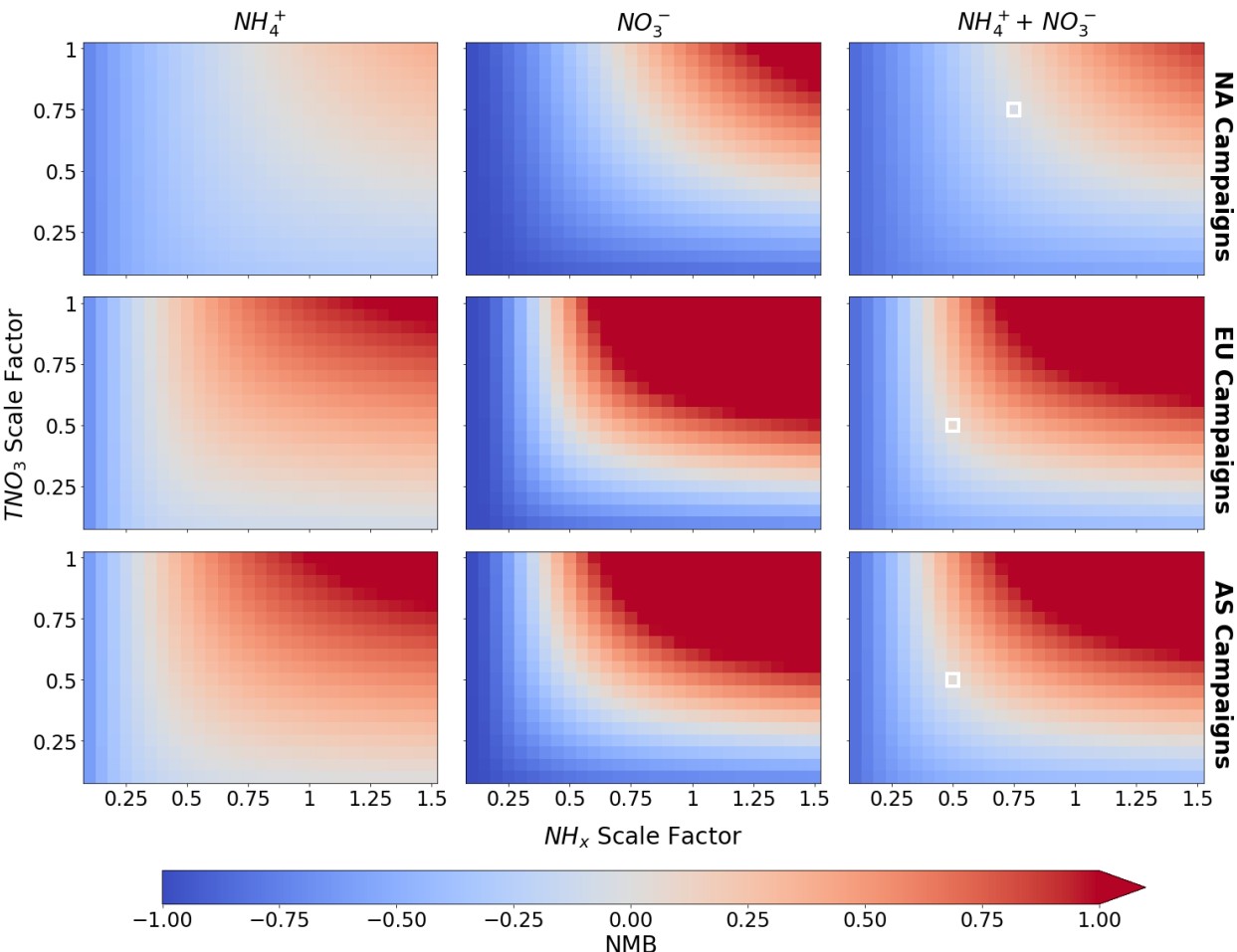

***Figure 12.*** *ISORROPIA II performance across different sensitivity runs conducted by scaling $NH_x$ and $TNO_3$ input from the baseline model values. Performance is reported as NMB for $NH_4^+$ (first column), $NO_3^-$ (middle column), and the sum of $NH_4^+$ and $NO_3^-$ (last column). Campaigns are grouped by region. Data is filtered to retain points where model T < 300 K and $15 \leq$ model RH < 90%. White boxes in the last column indicate the scaling factors for $NH_x$ and $TNO_3$ used in the full GEOS-Chem sensitivity test run (discussed in Sect. 5.2).*

## 5.2 Response of SNA to Changes in Emissions

Overestimated precursor emissions in the model could drive the high bias in ammonium nitrate in GEOS-Chem. We conduct a sensitivity test where we assume that the entirety of the ammonium nitrate model bias is associated with emissions uncertainties and use the concentration scalings for $TNO_3^-$ and $NH_x$ from the previous section as a proxy for $NO_x$ and $NH_3$ emissions in a GEOS-Chem sensitivity simulation. We cut both $NO_x$ and $NH_3$ anthropogenic emissions by 50% for the EU and AS regions and by 25% for the NA region (scalings for each region are highlighted by the white outlined boxes in Fig. 12). Agricultural emissions, which are included in the anthropogenic emission inventories in GEOS-Chem, are also scaled down. The reduction to anthropogenic emissions is performed as a simple sensitivity to the dominant source and does not imply that other smaller $NO_x$ sources (e.g. soil and lightning) are unbiased. The resulting GEOS-Chem model bias in nitrate and ammonium are both significantly

reduced (Fig. 13). This confirms that our offline ISORROPIA II sensitivity tests shown in Fig. 12 are a reasonable proxy for precursor emissions scaling. However, reductions in bias come without any significant improvement to the model's ability to capture the shape of the observed distribution or model skill (see $R^2$ values). In particular, despite the significant improvement at high $NO_3^-$ concentrations, the lower $NO_3^-$ concentrations (0.01 – 1µg/sm3) are still significantly underestimated suggesting that the biases at high and low concentrations may be driven by different factors. Furthermore, there is good agreement (within 10%) between the current $NH_3$ emissions from CEDS and a top-down satellite-based emission estimate for North America, Europe, and East China (Luo et al., 2022)., Also, a regional emissions inventory for Asia is within ±25% of $NO_x$ and $NH_3$ emissions estimates from CEDS (Kurokawa and Ohara, 2020).

In addition, for those campaigns where $NO_x$ was measured, the model is almost consistently biased low in $NO_x$ (NMB ~ -0.29) and overestimates the $HNO_3$:$NO_x$ concentration ratio (Fig. S14), which suggests that, rather than $NO_x$ emissions, formation (and loss) of $HNO_3$ may instead be overestimated (underestimated) in the model. Low $NO_x$ and high $HNO_3$ biases could also indicate that oxidation is too fast in the model. Overly rapid oxidation could also contribute to the high $SO_4^{2-}$/$SO_x$ ratios seen across some campaigns (Fig. S4). While we do not explicitly investigate the potential role of oxidation on SNA model bias, we note that the mean tropospheric OH burden in GEOS-Chem is on the higher end of what is suggested by the literature (based on both observations and models; Bloss et al., 2005; Hu et al., 2018). Direct comparisons of GEOS-Chem to observations made at surface sites and during aircraft campaigns show that modelled OH (including its uncertainty) generally falls within the uncertainty range of measured OH, but is generally higher in the model than the observations (Bloss et al., 2005; Christian et al., 2018; Kim et al., 2022). However, inconsistent biases in $HNO_3$ across the campaigns suggest that model OH is not exclusively driving model bias. As mentioned above, changes to VOC emissions can also affect SNA concentrations, leading to possible reductions in concentration and the model bias presented here (e.g., Aksoyoglu et al., 2017), however this effect is likely limited to near-surface regions with a higher potential for missing VOC reactivity and is unlikely to be an important driver of the high, consistent $NO_3^-$ bias seen here in the free troposphere.

While reductions to the emissions in GEOS-Chem can eliminate the bias in the model simulation, the poor (and worsening) model skill is not ameliorated, suggesting that regional emissions biases alone are not responsible for the poor model performance for SNA.

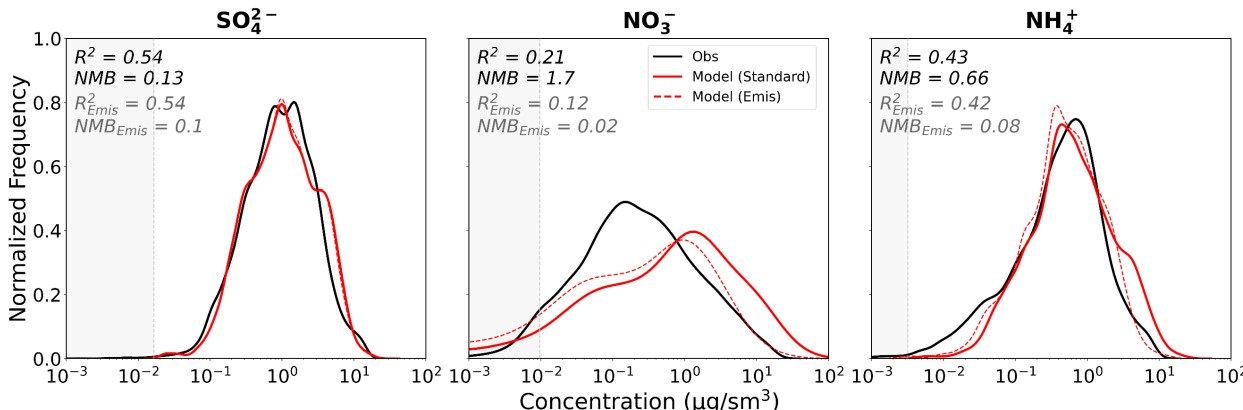

***Figure 13**. Distribution plots of sulfate, nitrate, and ammonium mass concentrations across the all campaigns as in the observations (black), the standard GEOS-Chem model run (red, solid), and the GEOS-Chem run with $NO_x$ and $NH_3$ emissions reduced ("Emis"; red, dashed). $R^2$ and NMB are reported for both the standard and reduced emissions simulations. Shaded regions indicate concentrations below the detection limit of the AMS (shown is median DL across all campaigns). Extreme T and RH values have been filtered as described in Sect. 5.1.2.*

### 5.3  Sensitivity of SNA to Dry Deposition Changes

Dry deposition of SNA and its precursors is not well constrained. Evaluation of current model parameterizations for dry deposition are limited by a relatively small number of direct global measurements available for dry deposition fluxes and large uncertainties in calculated deposition velocity ($v_d$; Emerson et al., 2020). Travis et al. (2022) suggest some of the high bias in GEOS-Chem's nitrate and $HNO_3$ during KORUS-AQ could be attributed to insufficient dry deposition on urban surfaces and see improvements in the model bias when $v_d$ for $HNO_3$ is increased by a factor of 5. Heald et al. (2012) saw weak responses of global surface nitrate concentrations (decreased by <10%) when $HNO_3$ dry deposition velocity was doubled.

Here we test how simulated global SNA responds to changes in $v_d$ using two sensitivity tests: one for changes in $v_d$ for all precursor gases ($SO_2$, $HNO_3$, and $NH_3$) and the other for changes in $v_d$ for all the SNA species. In both simulations, we increase $v_d$ by a factor of 2. We conduct these sensitivity tests for one year of simulation and not for all the campaigns (i.e. we do not provide comparisons of $R^2$ and NMB). Figure 14 shows that relative changes in surface concentrations are minimal across all species and the two different sensitivity tests. Over land, surface $NO_3^-$ is the most sensitive to the scaling of $v_{d,prec}$ and $v_{d,SNA}$. Scaling $v_{d,prec}$ has a larger effect on SNA concentrations than scaling $v_{d,SNA}$, demonstrating the more important role of dry deposition for the gas-phase precursors. However, while dry deposition of SNA in GEOS-Chem is on the lower end of other global models, dry deposition of precursors is on the higher end of these same models (Bian et al., 2017). Changing dry deposition velocities has a lessened impact aloft, especially for the sensitivity test where $v_{d,SNA}$ was doubled (e.g., at the 800mb level the maximum decrease for $NO_3^-$ is 20%), which confirms that the simulation of airborne measurements shown here is largely unaffected by uncertainties in dry deposition.

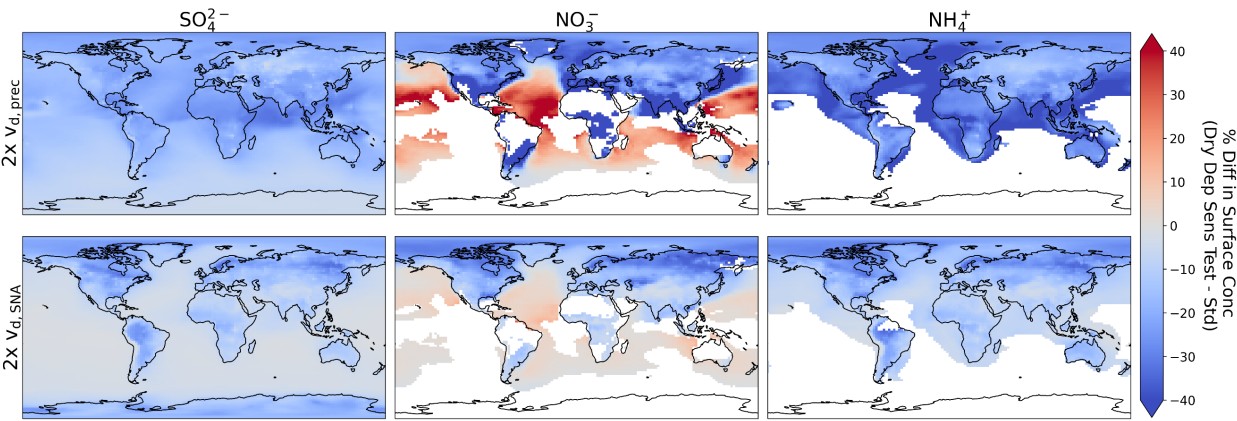

*Figure 14. Impact of doubling dry depositional velocity of precursor species (SO₂, HNO₃, and NH₃; top row) and SNA (bottom row) on annual mean surface concentrations of sulfate, nitrate, and ammonium for 2018. Concentrations < 0.05 µg/sm³ are filtered out.*

## 5.4 Sensitivity of SNA to Wet Deposition Changes

The wet deposition scheme in GEOS-Chem accounts for rainout and washout in both large-scale stratiform and convective precipitation as well as scavenging in convective updrafts (Jacob et al., 2000; Liu et al., 2001). These are highly parameterized processes that are empirically derived and remain uncertain. A recent update to the wet deposition scheme in GEOS-Chem was developed by Luo et al. (2019, 2020), including changes that are relevant to SNA concentrations. The Luo et al. scheme updated the value for in-cloud condensed water (ICCW) to vary temporally and spatially based on MERRA-2 cloud and rainwater, as opposed to being a constant value. It also includes updated empirical washout coefficients for $HNO_3$ and aerosols and rainout efficiencies for $HNO_3$ and $SO_2$ (Luo et al., 2019, 2020). Calculation of the effective Henry's law constant (H*) was updated to use a varying rain water pH (for washout) and cloud water pH (for rainout and scavenging in convective updrafts), as opposed to a constant value of 4.5. Calculations of H* were also updated for $SO_2$ and $NH_3$, specifically, with impacts on both wet and dry deposition (e.g., for the dry deposition scheme, the average $v_d$ is 0.8–1 times the value from the standard simulation). The global annual mean burden for sulfate, nitrate, and ammonium are reduced by 32%, 53%, and 37% under these changes in our 2018 simulation. $SO_2$ and $HNO_3$ global annual mean burdens decrease by 15% and 56%, respectively, in the simulation with the Luo et al. scheme. In contrast, the ammonia burden increases by 55% as a result of partitioning favoring gas-phase $TNH_x$ when $SO_4^{2-}$ and $TNO_3$ are reduced. We use the Luo et al. scheme to explore some of the sensitivity surrounding wet removal uncertainties through the lens of model performance for SNA.

Figure 15 shows the mass concentration distributions for all three SNA species across all campaigns for the observations and the two different wet deposition schemes. Despite the addition of a geographically varying ICCW, which we might expect to better represent the regional variability in wet removal, there is no significant improvement in the $R^2$. However, the new wet deposition scheme substantially reduces the nitrate NMB from 1.70 to 1.02. The comparison suggests that the shifted nitrate distribution overestimates the lower concentrations, however many of these concentrations may lie below the detection limit of the AMS and cannot be evaluated. The vertical profiles for nitrate show similar trends with shifts to lower concentrations at all altitudes, but no noticeable improvement in model performance compared to the profiles shown in Fig. 6 for the default model. The ammonium mass concentration distribution is also significantly shifted to lower concentrations which improves the NMB. A similar reduction is seen for the sulfate mass concentration distribution, but the displacement to lower concentrations (not seen in the observations) slightly worsens the overall NMB (from 0.13 to -0.16). This suggests that the Luo et al. scheme may overestimate wet removal of SNA. Dutta and Heald (2023) also show that the Luo et al. deposition scheme results in a substantial overestimate of observed nitrate wet deposition fluxes. This suggests that additional work is needed to optimize the removal efficiencies in GEOS-Chem considering the use of a physically varying ICCW. We note that smaller storms, which impacted some of the campaigns, may not be resolved at the resolution of the model, and therefore even with updates to the wet deposition scheme there is a limitation to how well the variability in wet removal

can be captured. Finally, these comparisons emphasize that wet removal plays a major role in controlling the lifetime and abundance of SNA; biases in the representation of these processes may explain some of the deficiencies in the simulation of model SNA concentrations.

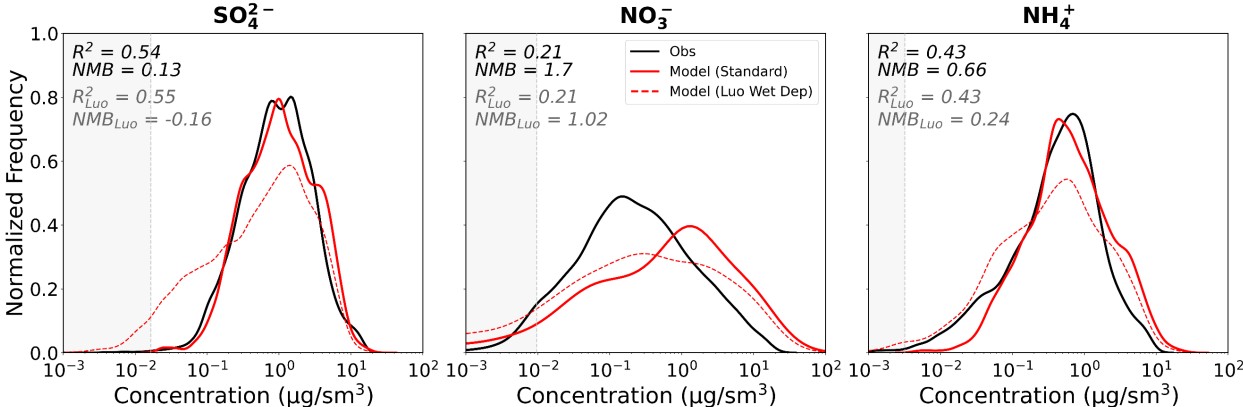

***Figure 15.*** *Distribution plots of sulfate, nitrate, and ammonium mass concentrations across all the campaigns as observed (black), simulated in the standard GEOS-Chem model (red, solid), and simulated in GEOS-Chem with the Luo et al. (2019, 2020) wet deposition scheme (red, dashed). $R^2$ and NMB are reported for both the standard (black text) and Luo et al. (grey text) simulations. Shaded regions indicate concentrations below the detection limit of the AMS (shown is median DL across all campaigns). Extreme T and RH values have been filtered as described in Sect. 5.1.2.*

### 5.5 The role of additional chemical sources and sinks in SNA bias

A missing chemical sink is another potential source of fine-mode SNA bias. Uptake of acidic gases (e.g., $HNO_3$, $SO_2$, $H_2SO_4$) by dust is one possible pathway. We find that for the two campaigns with the highest dust load (KORUS-AQ and EMeRGe-AS) acid uptake on dust, as implemented by Fairlie et al. (2010), improved the model's ability to capture SNA, but the impact was minimal. The largest impact was on nitrate where NMB was reduced by 0.04 and there was no change in model skill ($R^2$), consistent with previous results (Fairlie et al., 2010). Zhai et al. (2023) show that including anthropogenic coarse dust in GEOS-Chem eliminated much of the nitrate overestimate for the KORUS-AQ campaign observations made in the Seoul Metropolitan Area (SMA). In the SMA, the average coarse PM concentration at the surface was 23 µg/m³ for 2015 (Zhai et al., 2023), which is at the upper limit of what has been observed in Los Angeles and across European cities (range 5 – 23 µg/m³; Pakbin et al., 2010; Eeftens et al., 2012). Coarse anthropogenic PM is expected to be considerably less abundant outside of urban areas and aloft, and thus the campaigns explored here (including some individual flights during KORUS-AQ) would be relatively unaffected by this process, indicating that this is not a universal remedy for the GEOS-Chem nitrate simulation deficiencies.

Nitrate photolysis is another potential and uncertain pathway for nitrate loss. Studies generally relate the photolysis of nitrate to the photolysis of nitric acid by an enhancement factor (EF), with previous estimates for the EF ranging from 1 – 1000 (Romer et al., 2018; Shi et al., 2021; Ye et al., 2016). Shah et al. (2023) implemented a parameterization of $NO_3^-$ photolysis in GEOS-Chem to address an observed underestimate in NO, where the EF scales

from 10 to 100 depending on the concentration of sea salt aerosols relative to the concentration of $NO_3^-$. For two campaigns which are characterized by high calculated EFs and $NO_3^-$ concentrations (MILAGRO and WINTER, with mean EFs of 0.47 and 0.29 respectively), adding the Shah et al. scheme leaves $R^2$ unchanged and NMB negligibly altered ($\leq 0.02$) for all species. Therefore, nitrate photolysis, unless substantially more efficient than currently parameterized, cannot explain the large nitrate biases in the GEOS-Chem simulation.

We also consider the potential for an overestimated $HNO_3$ source to explain the SNA bias, specifically $N_2O_5$ uptake by aerosols. $N_2O_5$ hydrolysis represents a significant pathway for inorganic nitrate formation, estimated to contribute 41% of the inorganic nitrate source near the surface (Alexander et al., 2020) and 18% of the tropospheric inorganic nitrate burden (Alexander et al., 2009). The $N_2O_5$ uptake coefficient ($\gamma_{N2O5}$) indicates the probability that $N_2O_5$ will be lost on an aerosol surface, leading to the formation of $HNO_3$. The uptake parameter is dependent on

numerous factors (e.g., aerosol composition, temperature, RH) and there remains uncertainty in the model parameterization of this process, with estimated values ranging over several orders of magnitude (Holmes et al., 2019; Macintyre and Evans, 2010; McDuffie et al., 2018). In a sensitivity test, we reduced the uptake coefficient of $N_2O_5$ in GEOS-Chem by one order of magnitude across all aerosol types for the WINTER and KORUS-AQ campaigns, which have the highest concentrations of $N_2O_5$. There was no significant impact on $R^2$ ($\leq 0.01$) while the NMB for nitrate

for these campaigns was reduced from 1.90 to 1.72; this suggests that the uncertainty in this pathway has a limited, but non-negligible effect on the model's ability to capture SNA.

We explore the combined effect of all these updates to the chemical pathways (acid uptake by dust, reduced $\gamma_{N2O5}$, and $NO_3^-$ photolysis) on annual mean SNA. The global burden of both $SO_4^{2-}$ and $NH_4^+$ are negligibly impacted (~1% decrease), but there is a 11% reduction in the burden of $NO_3^-$. Fig. 16 shows that the largest impact on SNA

surface concentrations is for $NO_3^-$ over Eastern US, Europe, India, and East China. Sulfate concentrations show modest increases downwind of regions where $NO_3^-$ is decreased. A more damped effect on SNA concentrations is seen in the mid-troposphere. Collectively, known uncertainties in the chemical formation and loss processes (in the limits tested here) do not substantially perturb nitrate concentrations and cannot explain the model biases seen in our simulation.

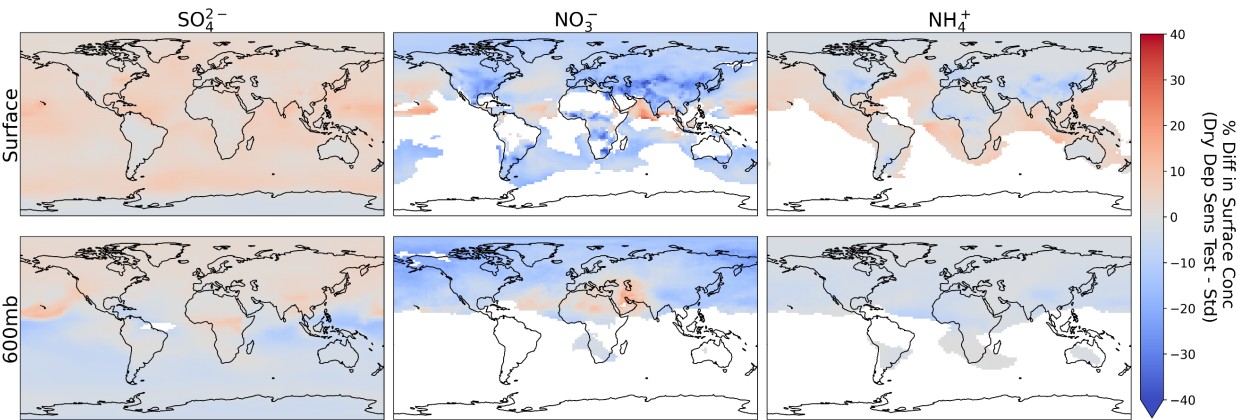

***Figure 16.*** *Impact of updates to chemical pathways in GEOS-Chem (i.e. including acid uptake on dust, $NO_3^-$ photolysis, and reducing $\gamma_{N2O5}$) on annual mean surface and mid-troposphere (600mb) concentrations of sulfate,*

*nitrate, and ammonium for 2018. Concentrations < 0.05 μg/sm³ are filtered out. Model nitrate and sulfate include nitrate and sulfate on dust in the smallest size bin.*

### 6 Conclusions

Our evaluation of the global inorganic aerosol simulation in GEOS-Chem against observations from 11 airborne campaigns indicates that sulfate is generally well simulated in the model but that there is a systematic high bias in nitrate (and ammonium), with worse performance in Europe and Asia. We explore a range of factors that may contribute to the bias in nitrate.

          We find that the ISORROPIA II model reproduces observed nitrate concentrations and conclude that
thermodynamic partitioning is not responsible for the model nitrate bias. However, we identify that the variability in observed $\varepsilon(NO_3^-)$ is not well captured with ISORROPIA II, but the evaluation of partitioning is incomplete given the limited set of ammonia observations. Extremely dry or saturated conditions, as well as the highest temperatures, are not well captured by ISORROPIA II, thereby degrading the GEOS-Chem model performance, particularly for nitrate. Removing these points modestly reduces the nitrate bias. Sensitivity tests using standalone ISORROPIA II suggest
that the model bias in other species ($HNO_3$, $Cl^-$, $Na^+$, $HCl$) are not responsible for the SNA bias. However, we find that partitioning is sensitive to $NH_3$ concentrations and, for the two campaigns with ammonia measurements, the model evaluation demonstrates little skill and significant biases for this species. Ammonia is not routinely measured; our results indicate that additional measurements are sorely needed to further explore how ammonia biases may impact model simulations of nitrate. With the caveat that the impact of a potentially poor ammonia simulation on nitrate
cannot be fully assessed, our analysis suggests that excessive sources or underestimated loss of nitrate in the model is the cause of the nitrate bias.

          The model is sensitive to adjustments in emissions, deposition, and, very minimally, to different chemical loss and production updates (i.e., acid uptake on dust, $N_2O_5$ uptake, and $NO_3^-$ photolysis), but none can explain the entirety of the high nitrate bias, or universally improve the model skill. Adjustments to the wet deposition scheme in
GEOS-Chem show reductions in nitrate bias but worsen the model's ability to capture sulfate, suggesting that nitrate concentrations are very sensitive to wet removal processes, but that these particular updates do not improve the model skill. A combination of changes to the emissions, deposition, and chemical production and loss may be able to close the high bias gap between model and observations, but more work is required to understand how to improve the model's ability to capture the variability in observed nitrate. We note that our comparisons assume that the fine-mode
SNA is fully captured by the AMS observations. A high model bias in nitrate may result if a substantial fraction of fine aerosol nitrate extends beyond the 1 μm size (and is mis-characterized by the model as sub-micron as well). Measurements of the aerosol nitrate size distribution extending up to 2.5 μm are needed to explore this further. More routine geographically distributed measurements of wet deposition of $TNO_3$ and dry deposition of $HNO_3$ may help also constrain the nitrate lifecycle. In addition, comprehensive measurements of $NO_y$ species (e.g., $N_2O_5$, PAN,

HONO, organic nitrates) would help to evaluate NO$_y$ cycling in the model and in turn identify how biases in the chemical processes involving NO$_y$ impact inorganic particulate nitrate.

The model deficiencies in SNA highlighted in this paper have broader implications because of the role of SNA in climate and air quality. Despite numerous updates over the past decade to the description of chemical and physical processes that are relevant to nitrate formation in GEOS-Chem, model predictions of nitrate concentrations
remain persistently biased high. The factor(s) contributing to the poor model skill and bias in SNA remain elusive. The grossly overestimated nitrate in GEOS-Chem implies that any policy-relevant studies for air-quality and climate that employ this model will be similarly biased, including an over-emphasis on nitrogen containing PM and a likely incorrect attribution of sectoral contributions to PM. Comprehensive measurements of particle and gas-phase precursors in a range of environments would be invaluable to future efforts to identify the drivers of nitrate bias and
to improve the fidelity of GEOS-Chem and possibly other models.

**Data Availability**

The data that support the findings of this study is available at https://doi.org/10.5281/zenodo.14029436. The GEOS-Chem data for version 13.3.4 is available at https://doi.org/10.5281/zenodo.5764874 (The International GEOS-Chem User Community, 2021). Observational data for MILAGRO (https://doi.org/10.5067/Aircraft/INTEXB/Aerosol-
TraceGas; INTEX-B Science Team, 2013), DC3 (https://doi.org/10.5067/Aircraft/DC3/DC8/Aerosol-TraceGas; DC3 Science Team, 2018), KORUS-AQ (https://doi.org/10.5067/Suborbital/KORUSAQ/DATA01; KORUS-AQ Science Team, 2019), and FIREX-AQ (https://doi.org/10.5067/SUBORBITAL/FIREXAQ2019/DATA001; FIREX-AQ Science Team, 2023) are available through NASA LaRC Data Archive. Data for CalNex (https://csl.noaa.gov/projects/calnex/; last access: 31 Oct. 2024) and SENEX (https://csl.noaa.gov/projects/senex/; last
access: 31 Oct. 2024) are available via the NOAA ESRL data archive. Observational data for WINTER (https://www.eol.ucar.edu/field_projects/winter; last access: 31 Oct. 2024) is available through NCAR EOL Archive. ADRIEX (https://data.ceda.ac.uk/badc/adriex; last access: 31 Oct. 2024) and EUCAARI (https://data.ceda.ac.uk/badc/faam/data/2008; last access: 31 Oct. 2024) data are archived at the Centre for Environmental Data Analysis (CEDA). Observational data for EMeRGe-EU (https://halo-db.pa.op.dlr.de/mission/95;
last access: 31 Oct. 2024) and EMeRGe-AS (https://halo-db.pa.op.dlr.de/mission/97; last access 31 Oct. 2024) are publicly available via the HALO database.

**Author contribution**

CLH and OGN designed the study. OGN performed the simulations and led the analysis. PCJ, HC, JLJ, KK, JL, AMM, BAN, JS, and AW provided inorganic aerosol measurements used in the analysis. SB, MNF, JRG, and JBN
provided gas phase measurements used in the analysis. OGN and CLH wrote and edited the paper with input from the co-authors.

**Competing Interests**

At least one of the (co-)authors is a member of the editorial board of Atmospheric Chemistry and Physics.

**Acknowledgements**

This work was supported by NOAA (grant no. NA19OAR4310180) and NSF (grant no. AGS-2223070). OGN was partially supported by the Rasmussen Fellowship, a graduate fellowship for the department of Earth, Atmospheric, and Planetary Sciences at MIT. PCJ, BAN, and JLJ were supported by NASA 80NSSC21K1451 and 80NSSC23K0828. BAN was also supported by NASA 80NSSC22K0283. We also acknowledge the following investigators who provided measurements for SNA, $HNO_3$, HCl, $Na^+$, $SO_2$, and CO: Andrew W. Rollins, J. Andy Neuman, Britton Stephens, Clifford Heizer, Felipe D. Lopez-Hilfiker, Glenn S. Diskin, L. Gregory Huey, Hans Schlager, Jack E. Dibb, Joel A. Thornton, John Crounse, John S. Holloway, Krystal T. Vasquez, Lisa Eirenschmalz, Lu Xu, Meghan Stell, Michael Lichtenstern, Michael Reeves, Michelle J. Kim, Paul O. Wennberg, Rodney J. Weber, Roya Bahreini, and Teresa Campos. Joel Thornton would like to acknowledge the NSF grant that funded WINTER: AGS-1360745.

The model simulations and analysis presented here were conducted using the "Svante" cluster, a facility located at MIT's Massachusetts Green High Performance Computing Center and jointly supported by the MIT Joint Program on the Science and Policy of Global Change; the Department of Earth, Atmospheric and Planetary Sciences; the Department of Civil and Environmental Engineering; the Institute for Data, Systems, and Society; and the Center for Global Change Science.

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
