# Peer review of "Exploring the processes controlling secondary inorganic aerosol: Evaluating the global GEOS-Chem simulation using a suite of aircraft campaigns"

_EGUsphere, 2024_

## Referee Comment (RC1)

**General Comments**

This paper looks at the performance of the GEOS-Chem chemical transport model across a range of field campaigns for secondary inorganic aerosols. This is an important topic as models generally struggle to simulate nitrate and ammonium in particular. The authors point out that this issue prevents successful policy-relevant studies for air quality and climate. Overall, the paper analysis is thorough, within the scope of ACP, and will hopefully help push the field to address this longstanding issue. The comments below seek to broaden the discussion and analysis in just a few ways. For example, the paper could be strengthened by adding evaluation of the sulfate and nitrate oxidation ratios for each campaign in a similar manner as Figure 4. As the paper already presents $SO_2$ and $HNO_3$ data, these ratios should be straightforward to include. A second general comment is that throughout the paper, statistics ($R^2$, NMB) are presented but it is unclear to which underlying data they apply – point by point aircraft vs. model data? Vertical profile averages? More clarity on this would help the reader understand whether model issues are temporal, spatial, or both. Finally, to best guide the community, the paper should very clearly lay out exactly what types of measurements are needed to improve these long-standing model biases, related to both production and loss processes.

**Specific Comments**

Line 42 – Why is nitrate partitioning difficult to assess due to limited ammonia? Do you mean total NHx? Can you also add a sentence to the abstract clarifying what measurements are needed to constrain nitrate in models?

Line 63 – Consider citing also (Zhai et al., 2021) here.

Line 67 – Not necessarily true in urban areas, non-agricultural contribution can be large, see these citations: (Chang et al., 2019; Lim et al., 2022; Link et al., 2017; Phan et al., 2013; Song et al., 2009; Sun et al., 2017)

Line 89 – You could also cite (Chen et al., 2019), they found that models tend to overly partition $HNO_3$ into nitrate.

Line 178 – There are many new developments related to aerosols in GEOS-Chem since 13.3.4. You comment on some of them later. It could help the reader to mention that you discuss recent developments such as nitrate photolysis in Section 5.5.

Figure 2 – Can you explain the longer lifetime for nitrate compared to ammonium or sulfate?

Line 265 – Does this poor $R^2$ reflect spatial and/or temporal issues? Is the diurnal cycle well captured but the campaign-to-campaign variability poor? Or is it both? It is unclear if the $R^2$ in Figure S1 is for the vertical variability?

Figure 4 – Can you put a unit on NMB? In the text on line 259 there is a unit of %, but it seems like the campaign-to-campaign biases for nitrate should be larger than a few %.

Line 327 – Why not run with E-AIM to compare to ISORROPIAII as has been done to compare against models by one of your co-authors (Nault et al., 2021)?

Line 370 – Is there a figure for the $NH_3$ improvement? Can you make a strong statement then that observations of $NH_3$ are key to future field campaigns?

Line 376 – Can you determine for which campaigns nitrate formation is limited by $NH_3$ availability (rather than NOx)? Is there a low bias in campaigns that are limited by $NH_3$?

Line 384 – Can you show model (not just offline ISORROPIA II) nitrate oxidation ratio just for the campaigns where you have $HNO_3$?

Section 5.1.2 – Figure 11 is very interesting. It might be sufficient to move Figure 10 to the supplement along with its associated discussion and just focus on Figure 11 here.

Line 483 – Again it would be useful to know whether observationally, ammonium nitrate for each campaign is limited by $HNO_3$ or $NH_3$.

Line 495 – Previously you stated that the main source of $NH_3$ was agriculture? If this is true in the model, shouldn't you cut agricultural sources not anthropogenic sources?

Line 534 – If Heald et al., 2012 saw a weak response from a doubling of $HNO_3$ dry deposition velocity, why would you expect a different response here in your similar sensitivity test? You cite a factor of 5 from Travis et al., 2022, but do not explore this possibility?

Line 535 – If 800 mb is roughly 1.5 km, that could be at the top of the boundary layer or higher for some campaigns. (like EMERGE-EU or WINTER). Maybe provide the value for ~900 mb as a better indicator of the boundary layer for all campaigns?

Line 552 – It would be clearer to say that this update to H* actually reduced wet and dry deposition.

Line 563 – Can you put your model sensitivity with the wet deposition changes on Figure 6 in a different color? Could you also put these model sensitivities on Fig. 5, Fig. S1, Fig. S2, and Fig. S3?

Line 600 – Why would FIREX-AQ have an EF of 0.71? Doesn't this mean that there was a high fraction of sea salt aerosol which doesn't make sense for an inland wildfire campaign?

Line 606 – Could you provide a budget for the other 59% of surface nitrate production? Is there any other pathway worth considering for a sensitivity test?

Line 621 – Models typically overestimate OH (e.g., (Prather and Zhu, 2024)). Do you think that model $HNO_3$ would be well simulated if model OH was correct? Or is it clear that the lifetime of $HNO_3$ is too long for some unknown reason?

Line 636 – Can you give the % of these observations in these extremes across all field campaigns? How much is a better thermodynamic model needed? Is the new HETP model expected to help with these extremes? (Miller et al., 2024).

Line 660 – What about wet and dry deposition? What measurements are needed to better understand those processes?

*References*

Chang, Y., Zou, Z., Zhang, Y., Deng, C., Hu, J., Shi, Z., Dore, A. J., and Collett, J. L.: Assessing Contributions of Agricultural and Nonagricultural Emissions to Atmospheric Ammonia in a Chinese Megacity, Environ. Sci. Technol., 53, 1822–1833, https://doi.org/10.1021/acs.est.8b05984, 2019.

Chen, L., Gao, Y., Zhang, M., Fu, J. S., Zhu, J., Liao, H., Li, J., Huang, K., Ge, B., Wang, X., Lam, Y. F., Lin, C.-Y., Itahashi, S., Nagashima, T., Kajino, M., Yamaji, K., Wang, Z., and Kurokawa, J.: MICS-Asia III: multi-model comparison and evaluation of aerosol over East Asia, Atmos. Chem. Phys., 19, 11911–11937, https://doi.org/10.5194/acp-19-11911-2019, 2019.

Jo, D. S., Emmons, L. K., Callaghan, P., Tilmes, S., Woo, J., Kim, Y., Kim, J., Granier, C., Soulié, A., Doumbia, T., Darras, S., Buchholz, R. R., Simpson, I. J., Blake, D. R., Wisthaler, A., Schroeder, J. R., Fried, A., and Kanaya, Y.: Comparison of Urban Air Quality Simulations During the KORUS-AQ Campaign With Regionally Refined Versus Global Uniform Grids in the Multi-Scale Infrastructure for Chemistry and Aerosols (MUSICA) Version 0, J Adv Model Earth Syst, 15, e2022MS003458, https://doi.org/10.1029/2022MS003458, 2023.

Lim, S., Hwang, J., Lee, M., Czimczik, C. I., Xu, X., and Savarino, J.: Robust Evidence of $^{14}$C, $^{13}$C, and $^{15}$N Analyses Indicating Fossil Fuel Sources for Total Carbon and Ammonium in Fine Aerosols in Seoul Megacity, Environ. Sci. Technol., acs.est.1c03903, https://doi.org/10.1021/acs.est.1c03903, 2022.

Link, M. F., Kim, J., Park, G., Lee, T., Park, T., Babar, Z. B., Sung, K., Kim, P., Kang, S., Kim, J. S., Choi, Y., Son, J., Lim, H.-J., and Farmer, D. K.: Elevated production of NH4 NO3 from the photochemical processing of vehicle exhaust: Implications for air quality in the Seoul Metropolitan Region, Atmospheric Environment, 156, 95–101, https://doi.org/10.1016/j.atmosenv.2017.02.031, 2017.

Miller, S. J., Makar, P. A., and Lee, C. J.: HETerogeneous vectorized or Parallel (HETPv1.0): an updated inorganic heterogeneous chemistry solver for the metastable-state $NH_4^+$–$Na^+$–$Ca^{2+}$–$K^+$–$Mg^{2+}$–$SO_4^{2-}$–$NO_3^-$–$Cl^-$–$H_2O$ system based on ISORROPIA II, Geoscientific Model Development, 17, 2197–2219, https://doi.org/10.5194/gmd-17-2197-2024, 2024.

Nault, B. A., Campuzano-Jost, P., Day, D. A., Jo, D. S., Schroder, J. C., Allen, H. M., Bahreini, R., Bian, H., Blake, D. R., Chin, M., Clegg, S. L., Colarco, P. R., Crounse, J. D., Cubison, M. J., DeCarlo, P. F., Dibb, J. E., Diskin, G. S., Hodzic, A., Hu, W., Katich, J. M., Kim, M. J., Kodros, J. K., Kupc, A., Lopez-Hilfiker, F. D., Marais, E. A., Middlebrook, A. M., Andrew Neuman, J., Nowak, J. B., Palm, B. B., Paulot, F., Pierce, J. R., Schill, G. P., Scheuer, E., Thornton, J. A., Tsigaridis, K., Wennberg, P. O., Williamson, C. J., and Jimenez, J. L.: Chemical transport models often underestimate inorganic aerosol acidity in remote regions of the atmosphere, Commun Earth Environ, 2, 93, https://doi.org/10.1038/s43247-021-00164-0, 2021.

Phan, N.-T., Kim, K.-H., Shon, Z.-H., Jeon, E.-C., Jung, K., and Kim, N.-J.: Analysis of ammonia variation in the urban atmosphere, Atmospheric Environment, 65, 177–185, https://doi.org/10.1016/j.atmosenv.2012.10.049, 2013.

Prather, M. J. and Zhu, L.: Resetting tropospheric OH and CH4 lifetime with ultraviolet H2 O absorption, Science, 385, 201–204, https://doi.org/10.1126/science.adn0415, 2024.

Song, C. H., Park, M. E., Lee, E. J., Lee, J. H., Lee, B. K., Lee, D. S., Kim, J., Han, J. S., Moon, K. J., and Kondo, Y.: Possible particulate nitrite formation and its atmospheric implications inferred from the observations in Seoul, Korea, Atmospheric Environment, 43, 2168–2173, https://doi.org/10.1016/j.atmosenv.2009.01.018, 2009.

Sun, K., Tao, L., Miller, D. J., Pan, D., Golston, L. M., Zondlo, M. A., Gri, R. J., Mauzerall, D. L., and Zhu, T.: Vehicle Emissions as an Important Urban Ammonia Source in the United States and China, Environmental Science, 51, 2472–2481, https://doi.org/10.1021/acs.est.6b02805, 2017.

Zhai, S., Jacob, D. J., Wang, X., Liu, Z., Wen, T., Shah, V., Li, K., Moch, J. M., Bates, K. H., Song, S., Shen, L., Zhang, Y., Luo, G., Yu, F., Sun, Y., Wang, L., Qi, M., Tao, J., Gui, K., Xu, H., Zhang, Q., Zhao, T., Wang, Y., Lee, H. C., Choi, H., and Liao, H.: Control of particulate nitrate air pollution in China, Nat. Geosci., https://doi.org/10.1038/s41561-021-00726-z, 2021.

---

## Author Comment (AC1)

We thank the reviewers for their helpful comments on our manuscript. Below, please find our responses in orange and the reviewers' comments in black. The line numbers in our responses refer to line numbers in the revised manuscript.

**Reviewer 1**

**General Comments**

This paper looks at the performance of the GEOS-Chem chemical transport model across a range of field campaigns for secondary inorganic aerosols. This is an important topic as models generally struggle to simulate nitrate and ammonium in particular. The authors point out that this issue prevents successful policy-relevant studies for air quality and climate. Overall, the paper analysis is thorough, within the scope of ACP, and will hopefully help push the field to address this longstanding issue. The comments below seek to broaden the discussion and analysis in just a few ways.

We thank the reviewer for their comments and suggestions.

For example, the paper could be strengthened by adding evaluation of the sulfate and nitrate oxidation ratios for each campaign in a similar manner as Figure 4. As the paper already presents SO2 and HNO3 data, these ratios should be straightforward to include.

It is not clear what the reviewer means by "sulfate and nitrate oxidation ratios". We infer from the second sentence that perhaps the reviewer would like to see the $SO_4^{2-}/SO_x$ and $NO_3^-/TNO_3$ ratios. We have added those vertical profile comparisons for the campaigns that include measurements of $SO_2$ and/or $HNO_3$ to the supplementary materials (Figures S4+6) and add the following text to the manuscript (line 293):

> "The ratio $SO_4^{2-}/SO_x$ (for campaigns that have $SO_2$ data) is well-captured for 4 of the 9 campaigns, but it is substantially overestimated for the remaining campaigns (CalNex, WINTER, MILAGRO, EMeRGe-EU, and EMeRGe-EU), particularly above the boundary layer (Fig. S4). For CalNex and MILAGRO, $SO_2$ is underestimated and $SO_4^{2-}$ is overestimated (while total SOx is well-captured), suggesting that oxidation may be overly rapid; for the other campaigns there is no evident relationship in the bias."

We also add a comment about $NO_3^-/TNO_3$ (line 447):

> "We also note that there is no systematic bias in the simulated $NO_3^-/TNO_3$ (Fig. S6). The over and underestimates in this ratio are consistent with the $NO_3^-$ bias seen in Figure 6, and thus are not indicative of a partitioning bias, further supporting the analysis of the previous section."

A second general comment is that throughout the paper, statistics (R2, NMB) are presented but it is unclear to which underlying data they apply – point by point aircraft vs. model data? Vertical profile averages? More clarity on this would help the reader understand whether model issues are temporal, spatial, or both.

The $R^2$ and NMB statistics presented on the vertical profiles have been changed to correspond to the vertical variability. We have clarified this in the captions of Figures 5-6 and Figures S3-8. Elsewhere,

$R^2$ and NMB apply to the point-to-point aircraft data, and we have added an explicit comment about this in the text (line 260):

> "These statistics are calculated for the point-by-point comparison between the observations and model or, only where explicitly mentioned, using the vertical profile.

Finally, to best guide the community, the paper should very clearly lay out exactly what types of measurements are needed to improve these long-standing model biases, related to both production and loss processes.

We agree with the reviewer that observational needs should be commented on here. Our submitted draft included a discussion of the critical need for ammonia measurements in the Conclusions. We have now expanded the discussion in our conclusions (line 712) to also highlight the importance of $NO_y$ measurements and how additional depositional measurements could assist in evaluating model biases (more on this below in response to another comment from this reviewer):

> "More routine geographically distributed measurements of wet deposition of $TNO_3$ and dry deposition of $HNO_3$ may help constrain the nitrate lifecycle. In addition, comprehensive measurements of $NO_y$ species (e.g., $N_2O_5$, PAN, HONO, organic nitrates) would help to evaluate $NO_y$ cycling in the model and in turn identify how biases in the chemical processes involving $NO_y$ impact inorganic particulate nitrate."

**Specific Comments**

Line 42 – Why is nitrate partitioning difficult to assess due to limited ammonia? Do you mean total NHx? Can you also add a sentence to the abstract clarifying what measurements are needed to constrain nitrate in models?

We have clarified in the abstract (line 43) that it is a complete set of gas and particle phase observations for SNA that are needed to constrain partitioning and that in particular ammonia is the critical component that is not often measured:

> "...but actual partitioning (i.e., $\varepsilon_{NO3} = NO_3^-/TNO_3$) is challenging to assess given the limited sets of full gas and particle phase observations needed for ISORROPIA II. In particular ammonia observations are not often included in aircraft campaigns and more routine measurements would help constrain sources of SNA model bias."

Line 63 – Consider citing also (Zhai et al., 2021) here.

The citation has been added.

Line 67 – Not necessarily true in urban areas, non-agricultural contribution can be large, see these citations: (Chang et al., 2019; Lim et al., 2022; Link et al., 2017; Phan et al., 2013; Song et al., 2009; Sun et al., 2017)

We agree and have adjusted the sentence to include this additional urban source of $NH_3$. Line 70 now reads:

> "The major sources of $NH_3$ are agricultural emissions, originating from livestock and fertilizer use, and, in urban areas, from vehicular emissions (e.g., Phan et al., 2013; Sun et al., 2017)."

Line 89 – You could also cite (Chen et al., 2019), they found that models tend to overly partition HNO3 into nitrate.

We have added the citation.

Line 178 – There are many new developments related to aerosols in GEOS-Chem since 13.3.4. You comment on some of them later. It could help the reader to mention that you discuss recent developments such as nitrate photolysis in Section 5.5.

We have added text in the model description section to refer to these processes not included in the standard v13.3.4 model (line 195):

"Acid uptake on dust (Fairlie et al., 2010) and nitrate photolysis (Shah et al., 2023) are optional processes in GEOS-Chem version 13.3.4 which we do not include in our model evaluation; however, we explore the effect of both of these processes on SNA in Section 5.5."

Figure 2 – Can you explain the longer lifetime for nitrate compared to ammonium or sulfate?

The wet removal efficiency and dry deposition rates are identical for all three of these species. Therefore, differences in lifetime largely reflect differences in the geographical and vertical distribution of where these aerosols are formed (see Figure 2). The ammonium lifetime is intermediate to sulfate and nitrate, reflecting its association with both of these ions.

Line 265 – Does this poor R2 reflect spatial and/or temporal issues? Is the diurnal cycle well captured but the campaign-to-campaign variability poor? Or is it both? It is unclear if the R2 in Figure S1 is for the vertical variability?

Given that aircraft moves in both space and time, it is not possible to separate the temporal and spatial variability. The aircraft also does not sample the full diurnal variability, but rather a snapshot of hours (typically daytime). Noted in response to the comment above, we have clarified in the main text and relevant captions what the $R^2$ and NMB statistics correspond to (i.e. vertical variability for the vertical profiles and point-to-point everywhere else).

Figure 4 – Can you put a unit on NMB? In the text on line 259 there is a unit of %, but it seems like the campaign-to-campaign biases for nitrate should be larger than a few %.

Thank you for pointing out this inconsistency. The NMB is a unitless ratio (as defined on line 257). We have removed any percentage units to ensure consistency throughout.

Line 327 – Why not run with E-AIM to compare to ISORROPIAII as has been done to compare against models by one of your co-authors (Nault et al., 2021)?

The primary goal of this work is not to investigate partitioning, but rather to test the GEOS-Chem model simulation of SNA. Given that GEOS-Chem uses ISORROPIA II, we briefly explore whether there are any obvious inconsistencies or errors introduced into GEOS-Chem by the use of this thermodynamic code. In section 5.1.1 we find that ISORROPIA II predicted $NO_3^-$ and $NH_4^+$ are relatively unbiased. We suggest that one reason for the imperfect offline ISORROPIA II performance is the lack of a complete observational dataset (and the use of substituted model values). Given this,

we cannot pursue any further investigation of the partitioning simulation, and a comparison with E-AIM would be subject to the same incompleteness.

Line 370 – Is there a figure for the NH3 improvement? Can you make a strong statement then that observations of NH3 are key to future field campaigns?

We have added figures for the two campaigns to SI (Figures S9 and S10). As discussed at length in the text, it is difficult to conclude robustly based on two campaigns that differ considerably, but we consider the lack of ammonia measurements to be the leading limitation for furthering our evaluation, as stated on lines 414 and 690.

Line 376 – Can you determine for which campaigns nitrate formation is limited by NH3 availability (rather than NOx)? Is there a low bias in campaigns that are limited by NH3?

That is a great question - we have added a comment regarding how $NH_3$ vs $HNO_3$ limitation relates to bias in the GEOS-Chem (line 278):

> "We examine if there is a connection between nitrate bias and the model gas ratio (Fig. S2), which is the ratio of free ammonia ($[NH_x]-2[SO_4^{2-}]$) to total gas + particle nitrate (Ansari and Pandis, 1998). A GR > 1 indicates that the system is $HNO_3$ limited, 0 < GR < 1 the system is $NH_3$ limited, and GR < 0 the system is extremely $NH_3$ limited and indicates that sulfate is not fully neutralized. When $NH_3$ is extremely limited, $NO_3^-$ concentrations are lower and there is consistent negative bias in the simulated $NO_3^-$. This suggests that GEOS-Chem has an excessively strong $NH_3$ limitation that is inhibiting some nitrate formation in these relatively clean (low SNA concentration) regions. However, these comparisons are also subject to measurement detection limits. The majority of the observations are characterized by GR > 0, which includes both ammonia limitation (0 < GR < 1) and $HNO_3$ limitation (GR > 1); the simulated nitrate is positively biased in both cases, which indicates that the model bias is not the result of one specific precursor limitation."

Line 384 – Can you show model (not just offline ISORROPIA II) nitrate oxidation ratio just for the campaigns where you have HNO3?

As mentioned above, we have added a figure (Fig. S6) and comment (line 447) about the ratio $NO_3^-$/$TNO_3$, specifically that it does not point towards a partitioning bias.

Section 5.1.2 – Figure 11 is very interesting. It might be sufficient to move Figure 10 to the supplement along with its associated discussion and just focus on Figure 11 here.

The reviewer makes a valid point; however, we choose to leave Figure 10 in the main text in anticipation that some readers may want to do a quick visual comparison between how much the model vs observed values plotted in Fig. 10 differ and how much changing between these model and observed values effects the partitioning (Fig. 11).

Line 483 – Again it would be useful to know whether observationally, ammonium nitrate for each campaign is limited by HNO3 or NH3.

See comment above about gas ratio.

Line 495 – Previously you stated that the main source of NH3 was agriculture? If this is true in the model, shouldn't you cut agricultural sources not anthropogenic sources?

Our sensitivity test also included cuts to agricultural sources. The anthropogenic emissions inventories we use in GEOS-Chem (CEDS and NEI) include agricultural sources. We clarify this on line 534:

> "Agricultural emissions, which are included in the anthropogenic emission inventories in GEOS-Chem, are also scaled down."

Line 534 – If Heald et al., 2012 saw a weak response from a doubling of HNO3 dry deposition velocity, why would you expect a different response here in your similar sensitivity test? You cite a factor of 5 from Travis et al., 2022, but do not explore this possibility?

We did not expect a different result but show it here for completeness. Travis et al. (2022) chose this factor of 5 to close the gap between observations and modelled $NO_3^-$ specific to the KORUS-AQ campaign with the justification that this captured loss to urban surfaces. Therefore, the factor of 5 is not widely applicable to all the campaigns we are looking at.

Line 535 – If 800 mb is roughly 1.5 km, that could be at the top of the boundary layer or higher for some campaigns (like EMERGE-EU or WINTER). Maybe provide the value for ~900 mb as a better indicator of the boundary layer for all campaigns?

The numbers are provided at 800 mb to precisely capture the free troposphere, not the boundary layer (which is reflected in the surface concentration shown in Figure 14).

Line 552 – It would be clearer to say that this update to H* actually reduced wet and dry deposition.

We chose to include a comment about $v_d$ as a quick quantitative example of how the changes to H* in Luo et al. specifically affect the dry deposition. To make it clearer, we have added a comment that this is specific to the dry deposition scheme (line 605):

> "Calculations of H* were also updated for $SO_2$ and $NH_3$, specifically, with impacts on both wet and dry deposition (e.g., for the dry deposition scheme, the average $v_d$ is 0.8–1 times the value from the standard simulation)."

It would not make sense to comment on the changes to wet and dry deposition in the context of changes to H* alone, since the net impact on dry deposition is by a combination of factors (e.g., other explicit changes to the deposition properties of $NH_3$ and $SO_2$, changes in partitioning because of changes in $NH_4^+$ and other relevant species). We do also add some text about the effect of the Luo et al. simulation on the burdens of $SO_2$, $HNO_3$, and $NH_3$ to better quantify the impact on these precursor species (line 608):

> "$SO_2$ and $HNO_3$ global annual mean burdens decrease by 15% and 56%, respectively, in the simulation with the Luo et al. scheme. In contrast, the ammonia burden increases by 55% as a result of partitioning favoring gas-phase $TNH_x$ when $SO_4^{2-}$ and $TNO_3$ are reduced."

Line 563 – Can you put your model sensitivity with the wet deposition changes on Figure 6 in a different color? Could you also put these model sensitivities on Fig. 5, Fig. S1, Fig. S2, and Fig. S3?

We find that the modified wet deposition scheme generally decreases concentrations, which has a mixed effect on the ability of the model to capture the vertical profile (some modest improvements, some worsening). Given this mixed impact and that adding extra lines can obscure the key comparisons, we choose to omit those lines on the figures. We note that the impact of this sensitivity on the vertical profile of nitrate is already noted in the text on line 617.

Line 600 – Why would FIREX-AQ have an EF of 0.71? Doesn't this mean that there was a high fraction of sea salt aerosol which doesn't make sense for an inland wildfire campaign?

The EF is calculated as EF = [SALA]/([SALA]*[NO3] (where [SALA] is the accumulation mode sea salt concentration). To make this clearer, line 654 has been changed to read:

> "...where the EF scales from 10 to 100 depending on the concentration of sea salt aerosols relative to the concentration of $NO_3^-$."

Thus, the high EF for FIREX-AQ is a result of the low $NO_3^-$ concentrations in the model and the sea salt concentration is indeed low for FIREX-AQ (mean [SALA] = 0.02μg/m$^3$). However, the reviewer raises a good point that because FIREX-AQ has lower concentrations of $NO_3^-$ it is not the ideal campaign for testing the effect of $NO_3^-$ photolysis. We therefore have adjusted this part of the discussion to be focused on two campaigns, WINTER and MILAGRO, which both have relatively high EFs and $NO_3^-$ concentration. Nitrate photolysis still has a negligible impact on the model performance for these two campaigns. The text now reads (line 655):

> "For two campaigns which are characterized by high calculated EFs and $NO_3^-$ concentrations (MILAGRO and WINTER, with mean EFs of 0.47 and 0.29 respectively), adding the Shah et al. scheme leaves $R^2$ unchanged and NMB negligibly altered (≤ 0.02) for all species."

Line 606 – Could you provide a budget for the other 59% of surface nitrate production? Is there any other pathway worth considering for a sensitivity test?

The other major source of inorganic $TNO_3$ is $NO_2+OH$, which according to Alexander et al. (2020) contributes 41% of near surface nitrate. We comment on the sensitivity of our simulations to this pathway in the next comment.

The other processes identified in Alexander et al. (2020) have a small impact on the global scale and/or locally important over the ocean or tropics, so unlikely to have a large impact on the simulations presented here. To develop our own budget for surface nitrate production would require a more detailed set of sensitivity simulations to examine $TNO_3$ production. This is beyond the scope of this work, and the subject of on-going work in our group.

Line 621 – Models typically overestimate OH (e.g., Prather and Zhu, 2024). Do you think that model HNO3 would be well simulated if model OH was correct? Or is it clear that the lifetime of HNO3 is too long for some unknown reason?

The reviewer raises an interesting point. The model does generally tend to underestimate $NO_x$ and overestimate $HNO_3$, which could point to overly rapid oxidation (see the added Fig. S14). However, the bias in $HNO_3$ is not consistent everywhere (including both over and underestimates of $HNO_3$ over the United States), suggesting that the performance is unlikely to be driven exclusively by high model OH. We add a comment to line 549 regarding the potential role of oxidation on model performance:

> "Low $NO_x$ and high $HNO_3$ biases could also indicate that oxidation is too fast in the model. Overly rapid oxidation could also contribute to the high $SO_4^{2-}/SO_x$ ratios seen across some campaigns (Fig. S4). While we do not explicitly investigate the potential role of oxidation on SNA model bias, we note that the mean tropospheric OH burden in GEOS-Chem is on the higher end of what is suggested by the literature (based on both observations and models; Bloss et al., 2005; Hu et al., 2018). Direct comparisons of GEOS-Chem to observations made at surface sites and during aircraft campaigns show that modelled OH (including its uncertainty) generally falls within the uncertainty range of measured OH, but is generally higher in the model than the observations (Bloss et al., 2005; Christian et al., 2018; Kim et al., 2022). However, inconsistent biases in $HNO_3$ across the campaigns suggest that model OH is not exclusively driving model bias."

Line 636 – Can you give the % of these observations in these extremes across all field campaigns? How much is a better thermodynamic model needed? Is the new HETP model expected to help with these extremes? (Miller et al., 2024).

20% of the observations are in these RH and T extremes (see line 462).

The HETP model is scientifically identical to ISORROPIA II but is rewritten to allow for more accurate root solving and faster processing time. Therefore, we do not expect to help with these extremes.

Identifying why there is such significant scatter in the partitioning (Fig. S11) is an important priority for future research, however, as shown (Figure 7) the thermodynamic model is at the very least able to reproduce the expected concentrations.

Line 660 – What about wet and dry deposition? What measurements are needed to better understand those processes?

The reviewer raises an interesting question. While widespread depositional networks would provide better constraints on removal, these observations are subject to some important limitations. Namely: wet deposition does not distinguish between particle and gas phase, and dry deposition is generally inferred from concentrations with assumed deposition velocities. Thus, they do not generally provide a sufficient constraint on modeled removal processes. A paper from our group (Dutta and Heald, 2023) discusses some of these limitations. We therefore feel that deposition measurements (as they are currently made) could be helpful but are not the top priority for investigating nitrate formation and loss. We have nevertheless added them to our list of needed measurements as follows (line 712):

> "More routine geographically distributed measurements of wet deposition of $TNO_3$ and dry deposition of $HNO_3$ may help constrain the nitrate lifecycle."
* * *
**Reviewer 2**

Norman et al perform a comparison of observed and modeled (GEOS-Chem) sulfate-nitrate-ammonium (SNA) aerosol in order to address a longstanding issue in models of significant discrepancies in nitrate and ammonia. This is important to address in light of the role that SNA plays

in PM2.5 abundance. Policies aimed at addressing PM2.5 pollution can best be made with an improved understanding of what controls their abundance. The authors perform a thorough comparison of the model with observations from 11 field campaigns in the US, Europe and east Asia with the goal of addressing SNA discrepancies in regions impacted by anthropogenic pollution. Similar to other model-observation comparisons, they find good model-obs agreement for sulfate, but the model generally overestimates nitrate and underestimates ammonia. They use GEOS-Chem and a stand-alone version of the aerosol thermodynamic model ISORROPIA to examine reasons for these model biases. They are able to run certain factors out (biases in transport, precipitation, thermodynamic partitioning of HNO3/NO3- and NH3/NH4+, dry deposition, chemistry) and find that uncertainties emissions and wet deposition play a larger role.

This paper is well written and scientifically sound and is thus suitable for publication in ACP. Their ruling out of processes that don't impact the model-obs discrepancies should help to move the science forward.

We thank the reviewer for their comments and suggestions.

I have two minor suggestions for improvement:

In the abstract, it is not immediately clear how the partitioning of HNO3/NO3- is hard to assess with limited ammonia observations. It seems like you would need observations of both HNO3 and NO3-, not ammonia. By the end of the paper is it more clear what you mean, so perhaps you should include more information on this in the abstract.

We thank the reviewer for the comment and have updated the text in the abstract to clarify (line 43):

> "...but actual partitioning (i.e., $\varepsilon_{NO3}$= $NO_3^-/TNO_3$) is challenging to assess given the limited sets of full gas and particle phase observations needed for ISORROPIA II. In particular, ammonia observations are not often included in aircraft campaigns and more routine measurements would help constrain sources of SNA model bias."

Section 3.1 is missing a description of the sinks of SNA aerosol. Is wet and dry deposition the only sink? If so, say so. Similarly, it is unclear in Table 2 if the lifetime is determined by wet and dry deposition or also by other processes. How was the lifetime calculated?

In our standard simulation, the only sink of SNA is deposition. The only other potential sinks are acid uptake by dust and nitrate photolysis which we discuss later in the paper as a sensitivity test.

The lifetime is determined by wet and dry deposition only and we have added that qualification to the caption of Table 2. Lifetime is calculated as the ratio of burden to global total deposition.
* * *
**Reviewer 3**

Norman et al. present an evaluation of GEOS-Chem model predictions of sulfate-nitrate-ammonium (SNA) for aircraft campaigns. Box modeling is used to investigate possible drivers of error and Figure 11 is a useful demonstration of what could drive error. The authors conclude total nitrate is likely overestimated which drivers overestimates in particulate nitrate.

We thank the reviewer for their comments and suggestions below.

General comment: While Figure 11 is very convincing in showing nitrate errors likely drive SNA errors, the lack of role for partitioning errors wasn't completely demonstrated.

We focus on the role of thermodynamic partitioning in Section 5.1.1. and show in Figure 7 that while the thermodynamic partitioning may contribute to some of the variability in $NO_3^-$ ($R^2$ = 0.79 for ISORROPIA II output compared to observations), that ISORROPIA II is not a major contributor to specifically the high bias in nitrate, as seen in GEOS-Chem.

Figure 6 shows pNO3- is underestimated below 4km for FIREX while Figure S2 shows HNO3 is overestimated indicating a partitioning error.

FIREX-AQ does have opposing biases for $NO_3^-$ (low) and $HNO_3$ (high) as seen in the newly added Fig. S14. To investigate whether this is directly associated with partitioning, we look at the observed vs modelled $NO_3^-$ partitioning ($NO_3^-/TNO_3$) for ISORROPIA II standalone (the same as Fig. S11 but for individual campaigns). $\varepsilon(NO_3^-)$ for FIREX-AQ is captured well and under-estimated ($R^2$ = 0.57 and NMB = -0.5), which means that ISORROPIA II is partitioning too much towards gas-phase for this campaign. However, there is no consistent trend we can discern in the partitioning performance across all the campaigns, indicating that this is not a systematic bias in ISORROPIA II and is likely associated with inputs into ISORROPIA II (e.g., biases in model $NH_3$ or observations).

Consider that errors in partitioning will affect lifetime due to different remove rates of HNO3 vs accumulation mode particle deposition (e.g., Nenes et al. 2021).

We thank the reviewer for pointing this out. We have adjusted the text on line 355 to read:

"Issues with partitioning, which can also act as a strong control on dry deposition and lifetime of total (gas- + particle-phase) nitrate ($TNO_3^-$) and ammonium (NHx = $NH_3$ + $NH_4^+$; Nenes et al., 2021), could contribute to the model SNA bias."

Recent work has shown treating SNA as non-equilibrium can reduce model bias (Rosanka et al. 2024).

Thank you for pointing out this interesting study. Rosanka et al. (2024) finds that treating SNA as non-equilibrium produces mixed results (increases and decreases in bias), suggesting that including this complexity would not uniformly improve our simulations. We have added the following text on this (line 431):

"Representation of non-equilibrium thermodynamics can introduce some improvement in model bias for SNA but can also worsen model performance (Rosanka et al., 2024), suggesting that the missing non-equilibrium process in this work is unlikely a large contributor to the model bias shown here."

A clearer demonstration that partitioning is not the issue and/or some investigation to bound the role is needed.

Our detailed investigation of partitioning suggests that this is not a dominant source of nitrate bias in our model, but as discussed extensively in the text, without a complete set of observations to thoroughly examine the simulation of partitioning, we cannot definitively rule out errors in this

simulated process. Unfortunately, we cannot constrain what we do not know, and thus there is no clear approach to bound the role of partitioning.

Consider a providing an HNO3 and NH3 budget in Table 2 as well as more information on total NO3 and total NHx in the main text.

Table 2 has been updated to include information about $HNO_3$, $NH_3$, and also $SO_2$. We have also added more references to $TNO_3$ throughout the paper (i.e., line 447, Fig. S6, Fig. S14).

Consider adding a figure that synthesizes across all the sensitivity simulations so they can be more easily compared in terms of relative impact and direction of changes.

Since some of our sensitivity tests are for full year simulations and others are for individual campaigns, and thus cannot be strictly compared, we do not to include a summary figure.

Specific comments:

1. Introduction: consider mentioning how VOCs can modulate nitrate abundance (Womack et al. 2019).

   We have added a comment to this effect (line 82):

   > "VOCs can also act as a local control on SNA concentrations since they are directly connected to oxidation capacity and also are involved in alternative loss pathways for nitrate radicals (Aksoyoglu et al., 2017, Womack et al., 2019)."

2. Page 2, near line 76: consider rewording to emphasize that ammonia isn't reacting stoichiometrically first with sulfate then second with nitrate as bisulfate is a common form of sulfate. What is meant by the term neutralize? pH 7? Note aerosols always have charge balance when H+ and OH- are considered.

   Thank you for this correction. We have reworded this statement (line 80) as follows:

   > "Ammonia reacts with both acidic sulfate aerosols (to form different salts, e.g., ammonium bisulfate, ammonium sulfate) and nitrate (to form particulate ammonium nitrate; Seinfeld and Pandis, 2016)."

3. Figure 1: Add years and/or months on the campaigns.

   This information is provided in Table 1. Unfortunately, adding this information to the figure substantially decreases legibility so we prefer to keep as is. We have added a reference to Table 1 in the Figure 1 caption so that the reader is directed to this information:

   > "Dates for each campaign are included in Table 1."

4. Figure 3: What is the current accuracy of PM2.5 and/or OA predictions for the 2018 period?

   We have added a sentence referencing some recent global GEOS-Chem model evaluations to the text (line 229):

"Surface PM$_{2.5}$ has been evaluated in GEOS-Chem previously and it is generally within 50% of the observations (Lee et al., 2017; Weagle et al., 2018; Zhai et al., 2021b)."

5. Figure 6: are nitrate measurements above 3km during CalNEX below the limit of detection? They seem below the 0.

A high proportion of the points are negative for CalNex (25%), especially at higher altitudes (looking at Figure 6, all bins above 3km have > 60% negative points). We have added a comment about this in the text (line 302):

"As indicated by the NO$_3^-$ vertical profile for CalNex, this campaign measured many negative NO$_3^-$ concentrations (25% of all points), especially at higher altitudes (greater than 3km all altitude bins have > 60% negative points). While we do not remove these points for any of our model-observation comparisons, we note that the bias would remain but be modestly decreased if points below the detection limit were removed from our analysis."

6. Clarify methods for Figure 7 and the box modeling. Is a forward ISORROPIA calculation always used? Consider renaming Figure 7 x-axis to "measured concentration".

As per the reviewer's suggestion, we have changed the x-axis labels of Fig. 7, S5, and S8-9 to read "Measured Concentration". We have adjusted the wording when introducing ISORROPIA II to clarify that we are using it in the forward mode and also to make it clearer what we are comparing in Figure 7 (line 361):

"The ability of ISORROPIA II to partition successfully can be evaluated by providing the observations as an input to a standalone version of ISORROPIA II (in forward mode) and comparing the predicted partitioning to the expected partitioning (i.e., the observations)."

7. Section 5.2: Did you consider how changes to VOC emissions may affect total nitrate?

We do not look into how changes in VOC emissions may affect SNA. In addition to discussing how VOCs can affect SNA (in response to the previous comment), we have added an additional line to further emphasize that this could play a role and clarify that we do not examine it here (line 558):

"As mentioned above, changes to VOC emissions can also affect SNA concentrations, leading to possible reductions in concentration and the model bias presented here (e.g., Aksoyoglu et al., 2017), however this effect is likely limited to near-surface regions with a higher potential for missing VOC reactivity and is unlikely to be an important driver of the high, consistent NO$_3^-$ bias seen here in the free troposphere."

**Reviewer 4**

Norman et al. present a comparison between aircraft-based measurements of sulfate, nitrate, and ammonium aerosol concentrations and GEOS-Chem simulations. They find that the model has more skill in reproducing sulfate observations than nitrate observations. Using various sensitivity tests, they identify several mechanisms that the model is sensitive to, but none are able to fully correct for differences between observations and measurements. The manuscript is well-written and thorough. I offer just a few specific points and questions below that may serve to improve an already high-quality work.

We thank the reviewer for their comments and suggestions below.

The authors could use the findings as a call to routinely monitor ammonia concentrations, especially in aircraft campaigns

In addition to the comment we make in the conclusions about needing more ammonia measurements (line 690), we have also updated the abstract to more clearly emphasize this (line 44):

> "In particular, ammonia observations are not often included in aircraft campaigns and more routine measurements would help constrain sources of SNA model bias."

140: is only sub-micron SNA captured by the AMS? Does this match the modeled size cutoff in Geos-chem?

We thank the reviewer for raising this point. The AMS measures only a fraction of the sub-micron mode mass of particles since the collection efficiency is highest for particles 70 – 500nm, and the transmission decreases for aerosols both larger and smaller than this range (Jayne et al., 2000). For our simulations in GEOS-Chem, the only size dependent process for SNA is dry deposition and for this the radius is assumed to be 0.17um (with hygroscopic growth can reach 0.68um at 99% humidity), thus, the model represents a similar size range as the observations. However, it is possible that part of the SNA size distribution exceeds the sub-micron mode measured by the AMS and that this is mis-represented in the model. We do not have measurements of the nitrate size distribution to explore this possibility. We add some text to acknowledge this possibility on line 709:

> "We note that our comparisons assume that the fine-mode SNA is fully captured by the AMS observations. A high model bias in nitrate may result if a substantial fraction of fine aerosol nitrate extends beyond the 1 μm size (and is mis-characterized by the model as sub-micron as well). Measurements of the aerosol nitrate size distribution extending up to 2.5 μm are needed to explore this further."

Table 2 (and elsewhere, e.g., line 618): the authors use the term "burden"—I believe replacing this with something more specific such as "concentration" would be more precise.

We choose to report the burden because we are interested in the total mass in the atmosphere (or in this case the troposphere), as opposed to reporting what is occurring at one specific level (e.g., the surface). This is especially the case since we are looking at aircraft campaigns that sample a large vertical range of the atmosphere. The burden is also a useful metric for comparison between GEOS-Chem and other chemical transport models. Figure 2 shows global maps of the concentrations at two different surface levels in addition to the burden shown in Table 2.

Figure 4 (& in the Discussion): it would be useful to test (and possibly present) R along with $R^2$ to identify any anti-correlation.

In this work, we only use $R^2$ as a metric to evaluate how the model compares to the observations such that we can use it as a proxy for the fraction of the variability captured by the model. In light of the reviewer's comment, we have verified that the R values are all positive (except for the R values for the $NO_3^-$ vertical profiles of two campaigns) and added text to this effect on line 261:

> "R values (not presented here) are all positive except for those corresponding to the $NO_3^-$ vertical profiles (discussed in detail below) of two campaigns (CalNex and SENEX), where the model and observations show opposite trends with height."

260: The authors mention scaling by the nitrate NMB. When this scaling is performed, how does it affect bias in modeled total PM2.5 at ground-based monitors? It may be useful to add a brief discussion about GEOS-Chem's performance at ground-based monitors to establish whether the comparisons with aircraft campaigns are representative of previous model evaluations.

We appreciate the reviewer's comment. A comparison to ground-based monitors is outside the scope of this work as we are already including model evaluation using an extensive observational dataset of aircraft campaigns. In the Introduction, we reference other works that have evaluated GEOS-Chem using surface data (e.g., Park et al. 2004, Heald et al., 2012, Zhang et al., 2012, Zhai et al., 2023, Travis et al., 2022). These other studies have generally shown similar biases, in nitrate especially. Our quick sensitivity test to show the effect on $PM_{2.5}$ is not meant to be taken as the suggested required decrease needed in $NO_3^-$ in models because, as we show, there are large differences on the local to regional scales. Thus, we present this as an illustration of the possible implication of this model bias.

In light of this comment, we have isolated the Northern Hemisphere (the region where the observations constrain nitrate concentrations in this work) in the now added figure for PM2.5 in the model (standard and scaled, Fig. S1) and provide a range of values for each of our key continental regions (line 274):

> "When nitrate is scaled down based on the NMB across all the campaigns (NMB = 1.76), average $PM_{2.5}$ concentrations across Northern Hemisphere land decrease by 3.4%, with maximum reductions of 25% in Eastern US and East Asia and 33% in Europe (Fig. S1 in the Supplement)."

Section 5.1.1: what effect is the missing dust cations expected to have on nitrate partitioning?

We mention on line 406 that including dust chemistry (i.e. dust cations effect on partitioning and acid uptake on dust) in a global atmospheric chemistry model has been shown to increase nitrate surface concentration by 21% (Karydis et al., 2016).

475: it would be helpful to have a theoretical reason for using the cumulative nitrate and ammonium NMB—this approach weights the biases of the two chemicals equally even though they make up different fractions of PM mass.

The reviewer makes a good point that accounting for the relative contribution of each individual species would be a more useful approach here. It is difficult to weigh the NMB based on the contribution of each species to PM mass since $NO_3^-$ and $NH_4^+$ contributions vary a lot (see Figure 1). So, to address this, we have adjusted Figure 12 to present the NMB of the sum of $NO_3^-$ and $NH_4^+$.